# Research on composite foundation rheology using the combined rheological element model

Jin Qu[1,2,3], Haitao Mao [2,3]*, Fan Wang[2], Sheng Tang[1], Xiaoju Wang[2,3]

1 College of Agricultural Engineering, Shanxi Agricultural University, Jinzhong, Shanxi, China, 2 College of Urban and Rural Construction, Shanxi Agricultural University, Jinzhong, Shanxi, China, 3 Shanxi Smart Water-Saving Technology Innovation Center, Taiyuan, Shanxi, China

* maohaitao1234@163.com

## Abstract

The safety and stability of the dam are affected by the rheology created by the deep soft soil layer following composite foundation reinforcement. Based on the classical combined element modeling, a new combined rheological element model is proposed in this paper. The consolidated rheological and fluid-solid coupling model is established by COMSOL to simulate the deep soft soil composite dam foundation. The results demonstrate that the model can better respond to the actual situation of stress, deformation, and seepage in each stage of the dam, and the error is within 5%. The strength and stiffness of every component of the dam are most affected by the consolidation rheology and fluid-solid coupling of the dam foundation during the impoundment phase, and the growth rate of each index gradually slows down with the extension of the running time. The rheology of the deep soft soil composite foundation is continuous. During the normal operation of the reservoir, the consolidation and settlement of the composite foundation are basically completed in 4–5 years, and the subsequent settlement changes are affected by the rheology for decades. The research results theoretically support the construction of safe and reliable system of deep soft soil composite foundation including clay core rockfill dam.

## 1. Introduction

The coordination of dam deformation is one of the most important concerns when dams are erected on deep soft soils, which are extensively dispersed throughout China's southeast coast, on both sides of inland waterways, and in lake areas [1,2]. Under sustained loading conditions, the gradual increase in effective stress within the soil matrix, coupled with the progressive development of viscous deformation among soil particles, may induce significant foundation settlement in dam structures, thereby posing substantial threats to engineering safety. More research is required to determine how soft soil consolidation rheology and the fluid-solid coupling effect affect dam stress, deformation, and seepage control.

**Data availability statement:** All relevant data are within the manuscript and its Supporting Information files.

**Funding:** This work was supported by Water Resources Technology Promotion and Application Projects of Shanxi Province [Grant No. 2025GM36 and 2025ZF02] (MHT), the Natural Science Foundation of Shanxi Province [Grant No. 202103021223132] (WXJ), the Natural Science Foundation for Young Scientists of Shanxi Province [Grant No. 202403021222105] (QJ), the Talent Introduction Program of Shanxi Province [Grant No. SXBYKY202010] (WXJ), the National Natural Science Foundation for Young Scientists of China [Grant No. 42207102] (WXJ), and the Doctoral Recruitment Program of Shanxi Agricultural University (Grant No. 2021BQ65) (WXJ).

**Competing interests:** The authors have declared that no competing interests exist.

Soft soil has a weak bearing capacity due to its low strength and large deformation. To increase the bearing capacity, it is frequently necessary to perform reinforcement treatment in the main structural area. This is typically done with cement mixing piles or vibrated rubble piles [3,4]. In deep soft soil foundation, the reinforcing piles typically create a composite foundation made up of underlying stratum and reinforced zone rather than penetrating through the soft soil layer of the foundation entirely [5–7]. Numerous academics have studied the characteristics of this type of composite foundation, treating the underlying stratum and the reinforced zone as a double-layer foundation. Zou et al.[8] and Yang et al.[9]discovered that the consolidation deformation of the underlying stratum develops slowly, while the consolidation deformation of the reinforced zone occurs more quickly during the application of the upper load. The underlying stratum is thought to be the source of post-construction settlement deformation, and the reinforced zone will deform along with the underlying stratum [10]. Consoli et al. [11] concluded that tensile cracks at the bottom of a two-layer system consisting of a compressible residual soil layer and compacted topsoil invariably initiate the progressive damage process in the top layer of compaction. Lang et al. [12] derived an analytical solution for the degree of consolidation by taking into account the difference in initial consolidation stresses in the underlying stratum when examining the consolidation performance of a rigid pile combination foundation. The composite foundation was divided into three components by Ma et al. [13]: the underlying stratum, the reinforced section, and the unreinforced section. The settling of each section was then calculated using a unit model structure and a layer-by-layer summation approach. Feng et al. [14] proposed a new semi-analytical scheme to analyze the consolidation behavior of composite strata with floating impermeable columns in unsaturated soils based on the equal strain hypothesis. However, existing studies predominantly focus on consolidation theory while neglecting the significant rheological effects of soft soils. As evidenced by research, the absence of structural and rheological characteristics in soft soil consolidation studies fundamentally compromises the accuracy of settlement deformation predictions [15,16]. Consequently, solely considering consolidation behavior provides an incomplete perspective for comprehensively understanding composite foundation performance.

The rheological behavior of soft soil is remarkably significant, exerting substantial influence on stability and settlement due to time-dependent mechanisms [17,18]. Soils and piles together exert load on the composite foundation reinforced zone. Composite foundations are often homogenized before calculation due to the complex nature of the pile-soil action, which is influenced by the material properties of both soils and piles as well as the foundation form, consolidation process, boundary conditions, and loading technique [19,20]. Tests conducted on the deformation following the reinforced zone's homogeneity reveal that the deformation can be approximated by the rheological effect of granular rock in time. Two primary processes contribute to the temporal deformation of the underlying stratum: the consolidation process, which comes after the superstatic pore water pressure dissipates, and the soil's compression and rheological process, which is brought on by viscosity [21]. The rheological mechanisms of reinforced zone exhibit significant distinctions from those of the underlying stratum. Accurate

characterization of the rheological behavior in distinct soil layers enables a more realistic representation of the settlement patterns in composite foundations. To address this, researchers have developed advanced constitutive models, including the fractional-order Merchant model [22,23], modified Burgers model [24], and fractional-order Kelvin model [25,26], for simulating the rheological response of soft soil composite foundations and predicting their non-uniform settlement and deformation [27,28]. Nevertheless, challenges persist in parameter identification following soil stratification, limiting the accuracy of long-term settlement predictions. Additionally, the coupled effects of seepage, stress, and rheology are often neglected, which substantially reduces the model's applicability to real-world engineering conditions.

In summary, this study proposes a novel layered modeling approach for soft soil composite foundations by conceptualizing the system as two distinct zones: the reinforced zone and the underlying stratum. Drawing on the theoretical framework of classical component combination models, the proposed method features a well-defined mathematical formulation and streamlined parameter calibration procedures. Furthermore, the model incorporates the coupled effects of seepage and stress under consolidation-induced rheological behavior, providing a robust theoretical foundation for analyzing the time-dependent deformation and long-term stability of deep soft soil composite foundations. In parallel, using the Dakai clay core rockfill dam project on soft soil composite foundation as an example, COMSOL-Multiphysics is used to establish the dam's calculation model at each stage of filling, impoundment, and operation. Combined with the actual observation data, the accuracy of the model is verified, and the calculation results of the consolidation model without considering rheology are compared, so as to comprehensively evaluate the influence of the rheology of the composite foundation on the dam body, the dam foundation and the main structure, and clarify the influence mechanism of the rheology of the soft soil composite foundation on the clay core rockfill dam.

## 2. Soft soil composite foundation rheological modeling

The necessity of using distinct rheological models for zoning is determined by the temporally disparate deformation characteristics of the underlying stratum and the composite foundation reinforced zone. The classical component combination modeling idea connects the relationship between the underlying stratum and the reinforced zone.

### 2.1. H-K rheological modeling in reinforced zone

Experimental studies have shown that the *H-K* model can approximate the deformation curve in time of the upper part of the composite foundation after the reinforced zone has homogenized. This model can simulate the elastic, creep (not infinite deformation), and relaxation (not zero stress) characteristics of the rock and soil mass [29]. An elastic element (*H*) with a modulus of elasticity of E and a kelvin (*K*) body connected in series make up the *H-K* model. The kelvin body is made up of a viscous element (*η*) with a coefficient of viscosity and an elastic element (*E_0*) with a modulus of elasticity connected in parallel. The stress-strain as a function of time under stress σ is given as follows [30]:

$$\varepsilon(t) = \frac{\sigma}{E} + \frac{\sigma}{E_0}\left(1 - e^{-\frac{E_0}{\eta}t}\right)$$

(1)

where $\sigma$ is the constant stress; $t$ is the loading time; $E$ and $E_0$ are divided into the initial elastic modulus and creep model elastic modulus; $\eta$ is the creep model viscosity coefficient.

### 2.2. Merchant model of fractional order derivatives in the underlying stratum

Integer differentiation is a widely used rheological model that describes the soil's stress-strain relationship over time and yields an integer differential-type constitutive equation. Nevertheless, empirical findings indicate that the constitutive relation of integer type cannot be well reconciled with experimental data on soft soil [31]. Based on the current research progress of soft soil rheology, the time effect of the stress-strain relationship of soft soil can be described by the fractional order derivative Merchant model [32], which essentially substitutes an Abel clay pot for the ideal Newton clay pot in the

kelvin body of the *H-K* model, as illustrated in Fig 1. This is based on the current state of research in soft soil rheology. The stress-strain relationship is given by [33]:

$$\sigma(t) \sim d^\alpha \varepsilon(t)/dt^\alpha \tag{2}$$

where $\alpha$ is the fractional order, $0 \le \alpha \le 1$. At $\alpha = 1$, the element represents an ideal clay pot, representing the Newtonian element, and at $\alpha = 0$, the element degenerates into an ideal spring, representing an elastomer.

The following represents the constitutive relation expression of the Abel clay pot under constant stress $\sigma$ [34]:

$$\sigma = E_1 \tau^\alpha D^\alpha [\varepsilon(t)] \tag{3}$$

where

$$D^\alpha f(t) = \frac{1}{\Gamma(1-\alpha)} \frac{d}{dt} \int_0^t \frac{f(t-\tau)}{\tau^\alpha} d\tau \tag{4}$$

where $E_1$ is the elastic modulus, $\tau = \eta/E$, $\eta$ is the Newtonian fluid viscosity coefficient, $D^\alpha$ is the Riemann-Liouville fractional-order differential operator, and $\Gamma()$ is the Gamma function.

The deformation of soft soil under constant load is manifested in two parts in time: consolidation and creep. The fractionally derivative Merchant model is used to describe the deformation characteristics of soft soil, as shown in Fig 2. The expression is as follows: [35]:

$$\begin{cases} \varepsilon = \varepsilon_0 + \varepsilon_1 \\ \sigma = E_0 \varepsilon_0 + E_1 \tau^\alpha s^\alpha \varepsilon_1 \\ \sigma_0 = E_0 \varepsilon_0 \end{cases} \tag{5}$$

where $s$ is a Laplace variable; $\sigma$ and $\varepsilon$ are the total stress and strain of the model, respectively; $\varepsilon_0$ and $\varepsilon_1$ are the strains corresponding to the spring element and the fractional-order Kelvin element, respectively; $E_0$ and $E_1$ are the modulus of elasticity in the spring element and the fractional-order Kelvin model, respectively.

A Laplace mathematical transformation of Eq (4) yields:

$$\begin{cases} \bar{\varepsilon} = \bar{\varepsilon}_0 + \bar{\varepsilon}_1 \\ \bar{\sigma} = E_0 \bar{\varepsilon}_0 + E_1 \tau^\alpha s^\alpha \bar{\varepsilon}_1 \\ \bar{\sigma}_0 = E_0 \bar{\varepsilon}_0 \end{cases} \tag{6}$$

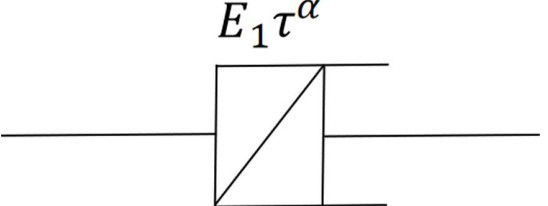

**Fig 1. Schematic diagram of Abel clay pot.**

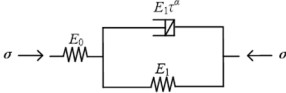

**Fig 2. Fractional order Merchant model.**

Combining the above equations gives:

$$\bar{\varepsilon} = \left( \frac{1}{E_0 + E_0 \tau^\alpha s^\alpha} + \frac{1}{E_1} \right) \frac{1}{s} \bar{\sigma}_0 \tag{7}$$

The Laplace transform of the flexible modulus of the fractional order derivative Merchant is:

$$J(t) = \left( \frac{1}{E_0 + E_0 \tau^\alpha s^\alpha} + \frac{1}{E_1} \right) \frac{1}{s} \tag{8}$$

A Laplace inverse transformation of the above equation yields:

$$J(t) = \frac{1}{E_1} - \frac{1}{E_0} \left\{ E_\alpha \left[ -\left(\frac{t}{\tau}\right)^\alpha \right] - 1 \right\} \tag{9}$$

where $E_a()$ is single parameter Mittag-Leffer function.

Thus, the fractional order derivative Merchant model stress-strain versus time is as follows:

$$\varepsilon(t) = \sigma \left\{ \frac{1}{E_1} - \frac{1}{E_0} \left\{ E_\alpha \left[ -\left(\frac{t}{\tau}\right)^\alpha \right] - 1 \right\} \right\} \tag{10}$$

Using Matlab, the experimental data can be fitted nonlinearly using least squares to determine the model parameters.

In addition to the model property parameters $E_0$, $E_1$, and $\eta$, the above two models are influenced by the material properties of Poisson's ratio $\mu$, permeability coefficient $k$, porosity $n$, etc.

### 2.3. Rheological model of soft soil composite foundation

Fig 3 illustrates the composite double-layer model of the dam foundation, which consists of two distinct structural components: the upper reinforced zone and the lower underlying stratum. The model incorporates impermeable and permeable boundary conditions at the base of the underlying stratum and the surface of the improved zone, respectively.

Therefore, from Eqs.(1) and (10), the stress-strain relationship of the dam foundation with time under the stress σ is obtained as follows:

$$\varepsilon(t) = \frac{\sigma}{E_0} + \frac{\sigma}{E_1}(1 - e^{-\frac{E_1}{\eta}t}) + \sigma \left\{ \frac{1}{E_0} - \frac{1}{E_1} \left\{ E_\alpha \left[ -\left(\frac{t}{\tau}\right)^\alpha \right] - 1 \right\} \right\} \tag{11}$$

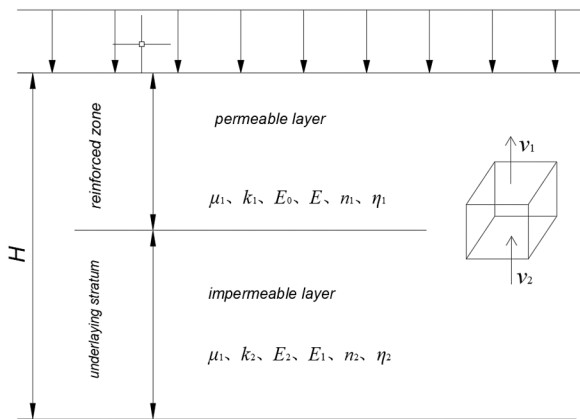

**Fig 3. "Double-layer" model of dam foundation.**

Boundary conditions are satisfied:

$$\nu_2 = \nu_1 \tag{12}$$

where, $v_1$ and $v_2$ are the seepage velocities of the reinforced zone and the underlying stratum at the contact surface, respectively.

## 3. Engineering overview

### 3.1. Dam overview

The clay core rockfill dam, spillway, diversion tunnel, pressure regulating well, plant, and other structures make up the primary structures of Dakai water conservancy. The location of the reservoir is shown in Fig 4. The clay core rockfill dam has a maximum height of 21 m, a length of 151 m, a top elevation of 1595.3 m, a top width of 8 m, and a bottom width of 147 m. The reservoir operates at a normal pool level of 1592.5 m, with no downstream water under normal operating conditions. The upstream slope of the dam, from the crest to elevation 1583.3 m, is designed at a gradient of 1:2.5, with an 18.5 m-wide toe berm at elevation 1583.3 m. Dam top to elevation 1580.3m has a downstream dam slope of 1:2, at elevation 1580.3m set 17m wide drainage prism, and the slope of the dam below the drainage prism is 1:2.5. A typical profile of the dam is shown in Fig 5.

### 3.2. Overlay characterization

The loose accumulation layer in the riverbed area has a simple structure and primarily split into three layers, based on the accumulation layer that the boreholes have revealed: ①The upper 0–9m is a sandy pebble gravel layer, which is loose and highly permeable, with permeability coefficients of $1.6 \times 10^{-2} \sim 4.46 \times 10^{-2}$ cm/s in general;②At an average depth of approximately 20 m, the middle- upper part is plastic gray-black and dark-gray clayey silt, which belongs to the high-liquid-limit clay with a permeability coefficient of $5.2 \times 10^{-5}$ cm/s; ③The middle and lower layers consist of hard plastic powdery clay that can be dug up to a maximum depth of 60 m. It has low compressibility, high strength, and low permeability, making it an excellent relative water insulation layer and foundation holding layer. The average permeability coefficient is $7.3 \times 10^{-6}$ cm/s.

### 3.3. Foundation treatment and cutoff wall arrangement

In accordance with foundation design specifications, the lower stiff plastic clay layer, exhibiting a standard bearing capacity of approximately 440 kPa, demonstrates low compressibility, high strength, and low permeability. These characteristics qualify it as both the bearing stratum and a relatively impervious layer for the dam foundation. The intermediate silty clay layer, with a standard bearing capacity approximately 190 kPa and moderate to high compressibility, constitutes the critical zone for deformation control in the dam foundation and serves as the primary target for ground improvement. The upper stratum, characterized by low density and high permeability, can be densified using vibro-compaction methods, supplemented by vibro-stone columns or cement-mixed piles to enhance bearing capacity. Accordingly, based on domestic dam foundation treatment practices, the vibro-stone column treatment method was adopted to reinforce the upper sandy gravel layer and intermediate plastic silty clay.

A zoned gradual arrangement is proposed to ensure a more uniform foundation settlement and improve the deformation and stress distribution of the dam body. This is done by taking engineering experience into account and adapting to the needs of different regions of the dam body for different requirements of foundation-bearing capacity. The stone columns have a diameter of 0.8 m, are pierced through the middle and upper layers of the dam foundation, and are spaced 1.5 m in the area of the larger dam height (the main body of the dam) and 2.0 m in the area of the smaller dam height (toe

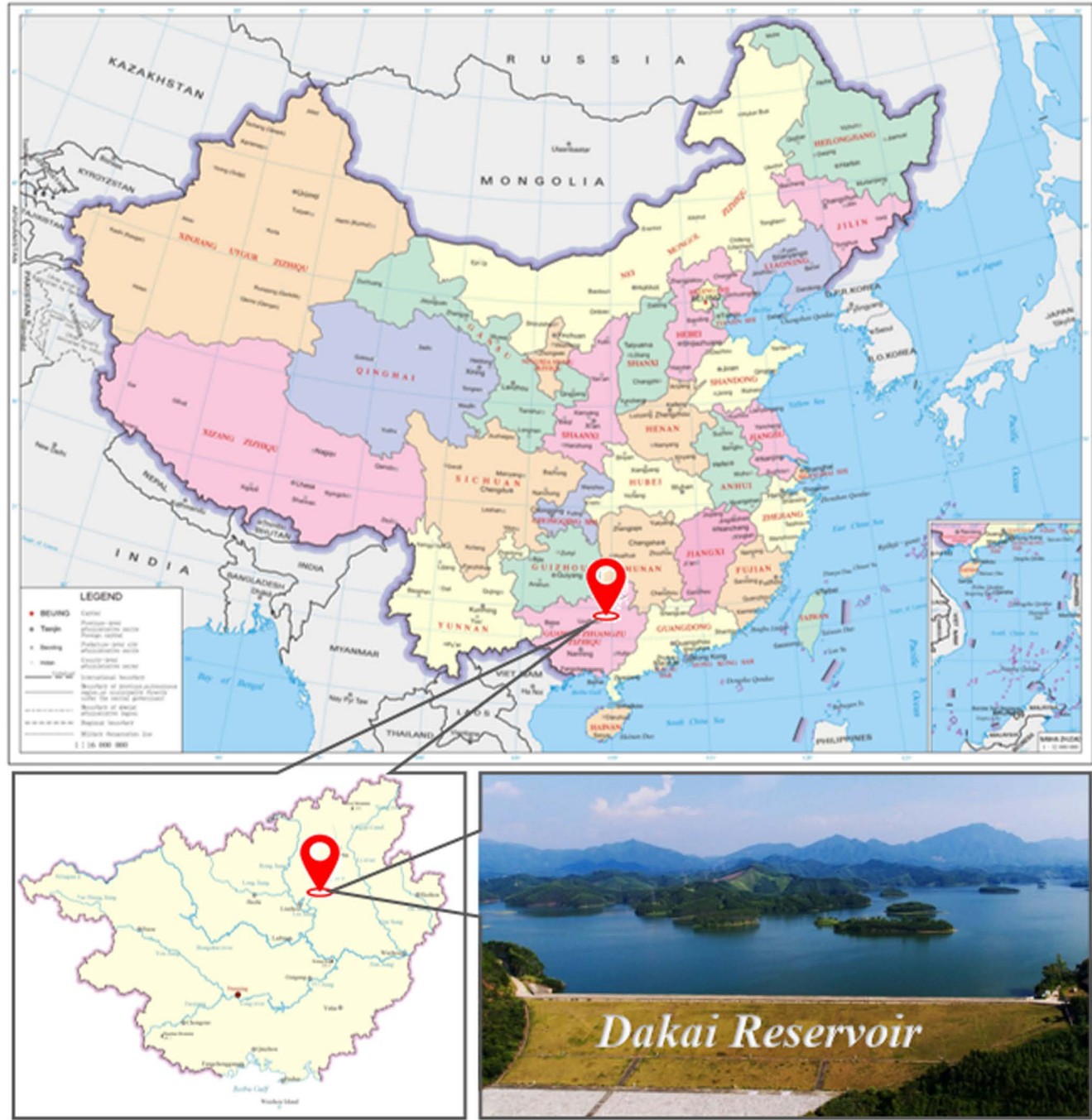

**Fig 4. Location of Dakai water conservancy project.**

berm). Combined with the permeability characteristics of each layer, the dam foundation adopts a suspended concrete cutoff wall with a wall thickness of 0.6m and a wall depth of 20m, which is embedded into the lower layer through the middle silty clay. The specific processing is shown in Fig 6.

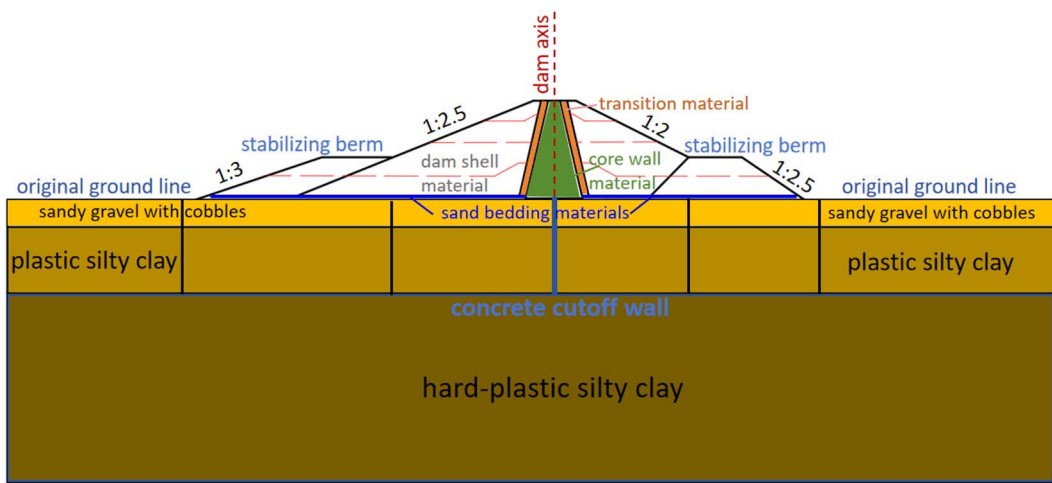

**Fig 5. Cross-section of the dam.**

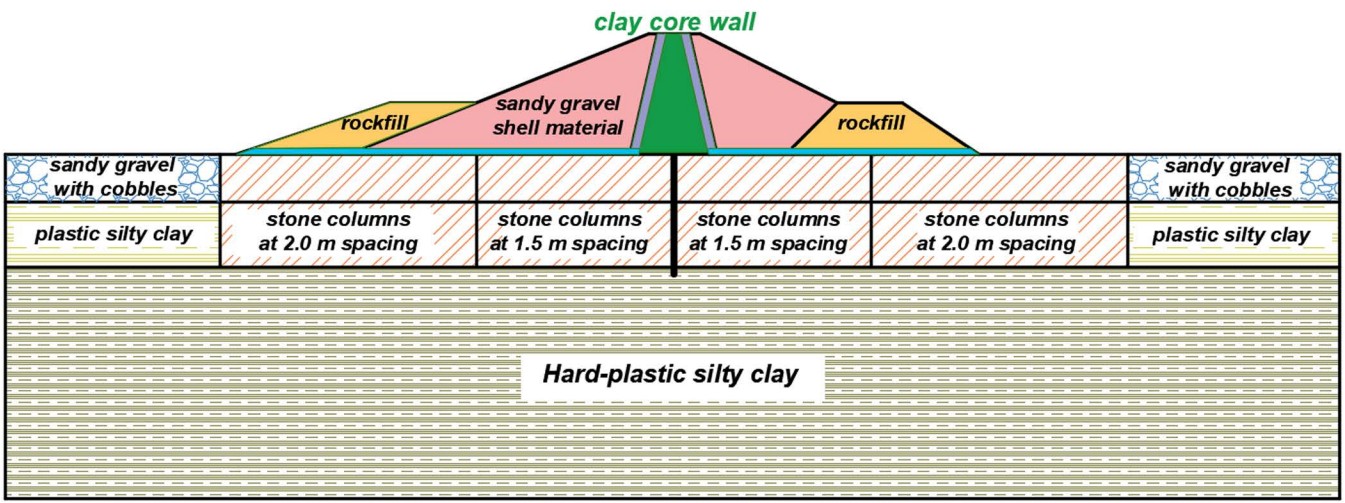

**Fig 6. Foundation treatment arrangement section.**

### 3.4. Monitoring system layout

The primary focus of safety monitoring for clay core rockfill dam during construction and operation mainly include:

① Dam surface deformation: install 20 groups of horizontal and vertical displacement monitoring points, spaced approximately 3 meters apart, and set up 4 longitudinal measuring lines at the upstream dam slope surface elevation of 10 and 18 meters and the downstream dam slope surface elevation of 17 and 12 meters.

② Internal deformation of the dam body:

a. Internal deformation of the clay core: electromagnetic settlement inclinometer tubes were employed for monitoring, with one inclinometer tube installed within the clay core at each cross-section. The base of each inclinometer tube was embedded to a certain depth below the dam foundation. A mobile inclinometer measures the horizontal displacement of

the clay core, while magnetic settlement plates are positioned at various heights outside the inclinometer tube and spaced 5 m apart to monitor the vertical displacement at that level.

b. Internal deformation of the dam shell rockfill: place a fixed inclinometer downstream of the clay core to track vertical displacement, and place a soil displacement meter downstream to track horizontal displacement. The standard distance between each measuring point for the horizontal fixed inclinometer is 6 m, and the distance between each measuring point for the soil deformation meter is 12 m.

③ Dam soil stress: six earth pressure transducer are buried in the upstream and downstream transition materials, and four are buried longitudinally at the dam's axis in the clay core, with a spacing of five meters.

④ Seepage monitoring of the dam base and body: install piezometer at the clay core's foundation surface, within the transition material's upstream and downstream foundation surfaces, and its upper portion to track the seepage pressure at various dam body heights. To keep an eye on the seepage pressure at the bases of the rockfill dams, a piezometer was placed at the base of the upstream and downstream dams, respectively. Seven piezometers are positioned longitudinally as a longitudinal observation section downstream of the cutoff wall in addition to the cross-section.

The specific layout of the monitoring system is shown in Fig 7. All the monitoring equipment of the dam is buried in the dam body when the dam is constructed, and then the signal lines and cables of the monitoring equipment are centrally led to the observation station on the top of the dam, and the subsequent monitoring is carried out in the observation station. When it is necessary to obtain monitoring data, the corresponding reading instrument is used to send electrical signals to the monitoring equipment, and the monitoring data can be obtained after processing the returned electrical signals. Fig 8 shows the monitoring equipment buried inside the dam body of Dakai Reservoir and the monitoring data acquisition process.

## 4. Numerical modeling

### 4.1. Rheological and consolidation coupling model

It is impossible to overlook how seepage affects the structure while impoundment and operational period. Based on flow theory in porous media, COMSOL-Mulphytics is used in this work to realize the coupled seepage stress (combined law

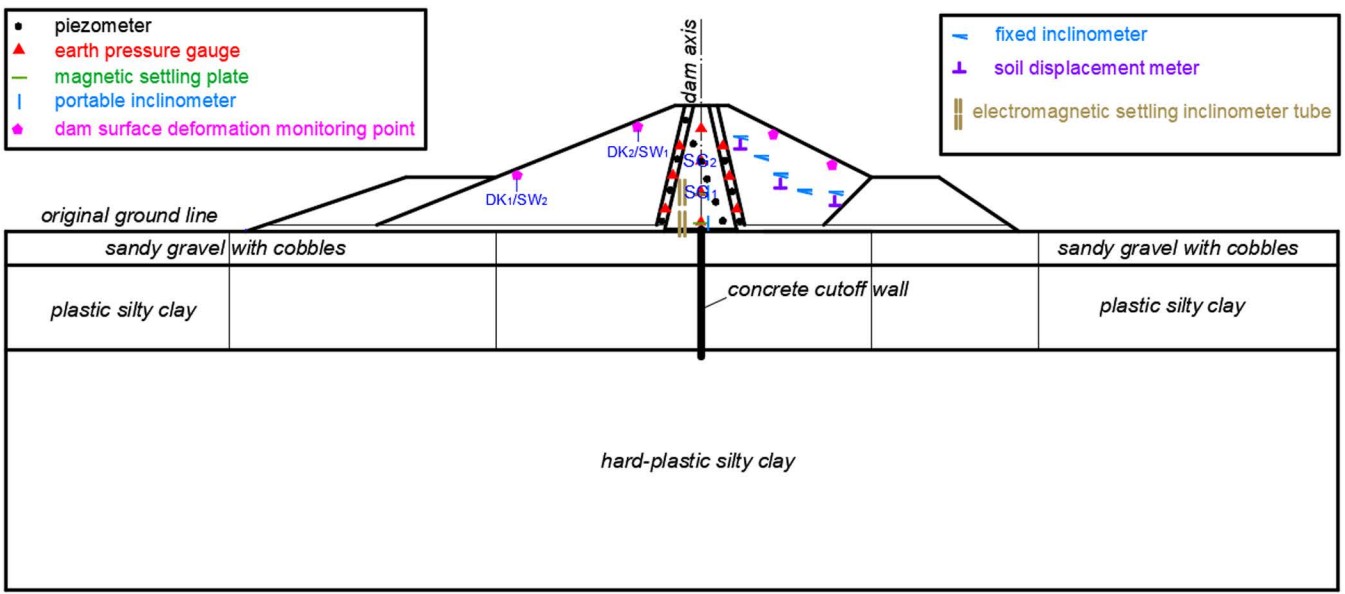

**Fig 7. Foundation treatment arrangement section.**

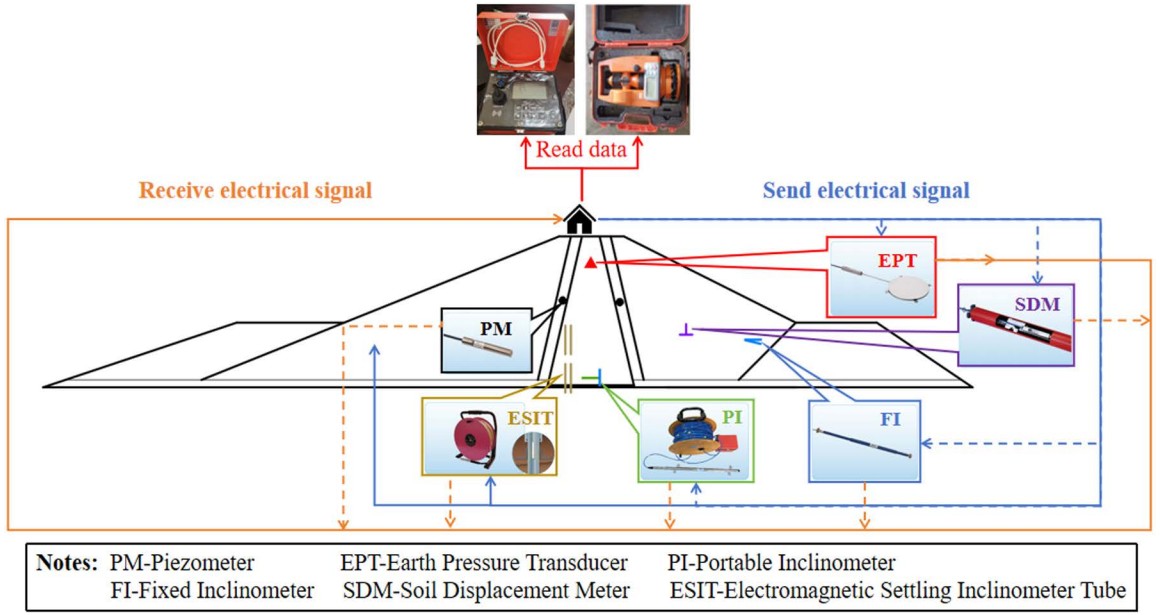

**Fig 8. Flow chart of dam monitoring data acquisition.**

of conservation of mass and Darcy's law) computations. Richard's equation, which describes the saturated-unsaturated coupling, can be used to predict the process of the dam material from initial unsaturated to gradually saturated as the reservoir level rises over the impoundment period. Given that the dam foundation lies below the surface of the water table, it can be assumed that the dam foundation material is saturated during the impoundment and normal operation period. This process can be modeled using porous elastic storage describing saturated coupling.

### 4.2. Model governing equation

**4.2.1. Richard coupling model for dams.** The model's governing equation for fluid continuity is [36,37]:

$$\rho\left(\frac{C_m}{\rho g} + S_e S\right)\frac{\partial p}{\partial t} + \nabla\rho\left(-\frac{k_s}{\eta}k_r(\nabla p + \rho g \nabla H)\right) = Q_m \tag{13}$$

where $\rho$ is the fluid density, $C_m$ is the water capacity, g is the gravitational acceleration, $S_e$ is the saturation, $S$ is the storage coefficient, $p$ is the pressure, $k_s$ is the saturated permeability, $\eta$ is the fluid viscosity, $k_r$ is the relative permeability, $H$ is the position head, and $Q_m$ is the fluid source-sink term.

The steady-state equilibrium governing equations of solid mechanics in the model based on Biot consolidation theory is [38]:

$$\begin{cases} \nabla \cdot (S + S_{ext}) + P_f = 0 \\ S_{ext} = -S_e(p - p_{ref})I \end{cases} \tag{14}$$

where $\nabla$ is the gradient operator, $S$ is the water storage model, $S_{ext}$ is the saturation coefficient, $P_f$ is the pore pressure, $S_e$ is the degree of saturation, $p$ is the absolute pressure, and $p_{ref}$ is the reference pressure level.

**4.2.2 Modeling of porous elastic storage at the dam base.** The fluid continuity governing equation is as follows [39]:

$$\rho S \frac{\partial H_t}{\partial t} - \nabla\rho\left[\frac{k}{\mu}(\nabla P_f + \rho g \nabla H)\right] = -\rho\alpha_B\frac{\partial}{\partial t}\varepsilon_{vol} \tag{15}$$

where $\rho$ is the liquid density, $S$ is the storage coefficient of the porous medium, $H_t$ is the total head; $P_f$ is the pore pressure, $k$ is the permeability coefficient, $\mu$ is the liquid kinetic viscosity, $g$ is the gravitational acceleration, $H$ is the positional head, $\alpha_B$ is the Bio-consolidation parameter, and $\varepsilon_{vol}$ is the porous medium body strain.

The pore deformation equation for porous media is [40]:

$$\begin{cases} \boldsymbol{\sigma} = \mathbf{C}\varepsilon - \alpha_B P_f \boldsymbol{I} \\ P_f = M(\zeta - \alpha_B \varepsilon_{vol}) \\ P_m = -K_s \varepsilon_{vol} + \alpha_B P_f \end{cases}$$

(16)

where $\boldsymbol{\sigma}$ is the Cauchy stress tensor, $\mathbf{C}$ is the elasticity matrix, $\boldsymbol{\varepsilon}$ is the strain tensor, $\alpha_B$ is the coefficient of specific occlusion, $P_f$ is the fluid pore pressure, $\boldsymbol{I}$ is the unit matrix, $\zeta$ is the value of the variation of the fluid content in the porous medium, $P_m$ is the volume change of the soil particles, and $K_s$ is the elastic modulus of the solid.

## 4.3. Model parameters

**4.3.1. Parameter value.** The Poisson ratio ($\mu$), permeability coefficient ($k$), foundation bearing capacity parameter ($\beta$), bulk density ($\gamma$), and porosity ($n$) of each rock and soil layer are obtained through a combination of experience value and indoor and outdoor experiments [41], in accordance with the concrete engineering characteristics of the composite foundation of Dakai clay core rockfill dam. The initial elastic modulus $E$ and $E_1$, the elastic modulus $E_0$ and $E_2$ of the creep model, and the viscosity coefficient $\eta$ of the creep model of the dam foundation are obtained through inversion calculation by utilizing the mathematical formula of the improved rheological model and combining with the settlement monitoring data of the dam body during the filling period. The calculated parameters of the rheological model of the dam foundation are shown in Table 1.

Since the material used for the dam body is nonlinear, the finite element calculation uses the Duncan $E$-$\upsilon$ model, which has fewer parameters, more test data, and a better reaction to the material's nonlinear features. Table 2 displays the parameters that were utilized in the computation.

Furthermore, a linear elastic model was employed for the concrete cutoff wall, whereby the density, modulus of elasticity, Poisson's ratio, and permeability coefficient were determined to be 2.4g/cm³, 2.8GPa, 0.167 and $1 \times 10^{-12}$m/s, respectively.

**4.3.2. Parameter verification.** The inversion parameter were compared with the rheological experimental parameters of soil and stone in each layer, as indicated in Table 3, to confirm the rationality of the creep parameters in the reinforced zone and the underlying stratum and guarantee the accuracy of the numerical model calculation results. Table 3 shows that the variation rules of the inversion parameters and experimental parameters are consistent, with the error between them being less than 10%. There is no variation in the order of magnitude of the same parameter, indicating that the model parameters derived from the inversion calculation are feasible.

## 4.4. Modeling

The composite rheological model of soft soil and the fluid-solid coupling model are combined by COMSOL to establish the composite rheological model of soft soil and the fluid-solid coupling model. First, using the step-by-step loading method,

**Table 1. Parameters of composite foundations for Dakai clay core rockfill dam.**

| blanket | $\gamma$ | modulus of elasticity/MPa | | | | $\beta$ | $\eta$ | $\mu$ | $k$ | $n$ |
|---|---|---|---|---|---|---|---|---|---|---|
| | kN/m³ | $E$ | $E_0$ | $E1$ | $E_2$ | | $\times 10^{17}$ | | $\times 10^{-7}$ cm/s | |
| reinforced zone | 18.5-19.0 | 30.23 | 12.7 | – | – | 1.0 | 13.0 | 0.43 | 3.5 | 0.23 |
| underlying stratum | 16.0 | – | – | 20.05 | 7.6 | 0.36-0.50 | 1.2 | 0.4 | 7.33 | 0.3 |

**Table 2. Parameters of *E*-υmodel for Dakai clay core rockfill dam material.**

| materials | γ kN/m³ | c/kPa | $\varphi_0$ | K | ns | $R_f$ | G | F | D | Δφ | Kur |
|---|---|---|---|---|---|---|---|---|---|---|---|
| clay core | 19.1 | 60.0 | 20 | 300 | 0.50 | 0.75 | 0.35 | 0.08 | 4.0 | 0 | 500 |
| transition material | 20.5 | 0.0 | 48 | 750 | 0.45 | 0.72 | 0.46 | 0.26 | 5.0 | 8 | 1300 |
| dam shell material | 22.0 | 0.0 | 50 | 800 | 0.42 | 0.72 | 0.48 | 0.26 | 5.5 | 8.5 | 1500 |
| rock fill | 22.5 | 0.0 | 54 | 720 | 0.303 | 0.80 | 0.44 | 0.20 | 5.2 | 13.5 | 1600 |
| gravel layer | 22.0 | 0.0 | 51 | 1050 | 0.354 | 0.71 | 0.43 | 0.24 | 5.0 | 7 | 2210 |

**Table 3. Comparison of parameter values.**

| blanket | inversion parameter | | | | | experimental parameter | | | | |
|---|---|---|---|---|---|---|---|---|---|---|
| | E | E0 | E1 | E2 | η ×1017 | E | E0 | E1 | E2 | η ×1017 |
| reinforcement layer | 30.23 | 12.7 | | | 13.0 | 31.71 | 13.56 | | | 12.37 |
| underlying stratum | | | 20.05 | 7.6 | 1.2 | | | 21.78 | 8.69 | 1.11 |

the dam body is filled four times, with a single application of each filling load, following the initial ground stress balancing of the soft soil composite foundation. Through the pre-stress and pre-strain parts of the solid mechanics module in the software, the stress-strain determined by each applied load is passed to the loading process of the subsequent stage, and the dam construction process is finished one step at a time. Second, the stress-strain value following the construction of the dam is taken as the initial value of the water storage stage. The software sets the water-time function ($h = 75 + 0.5t$) to achieve a uniform water storage of 0.5m per day to complete the water storage process. Thirdly, the stress-strain value after the completion of water storage is considered as the initial value of the dam operating stage. Two analysis modes of dynamic calculation and steady-state calculation are set by software to compare the time necessary for stable operation. A flow diagram of the model creation is shown in Fig 9.

Three months is suggested as the duration for each step of dam filling in the overall model calculation. The initial water storage is evenly performed at 0.5m/d for 36 days. This process will take approximately ten years following stable operation, after the completion of water storage, regardless of changes in water level throughout operation. Table 4 displays the corresponding time for each working situation. Fig 10 displays the model's grid division.

#### 4.5. Model verification

**4.5.1. Verification of displacement.** In order to verify the accuracy of the calculation results, combined with the layout of displacement monitoring points at the dam slope, the measured values of horizontal displacement (along the river) measuring points $DK_1$ and $DK_2$, settlement displacement measuring points $SW_1$ and $SW_2$ were selected and compared with the model calculation results. Fig 11 shows the displacement changes over time at the four measuring points obtained in COMSOL. The comparison between the measured values and the calculated values of the model is shown in Table 5 and Table 6.

Table 5 shows that there is a 10% margin of error between the measured and computed values for the horizontal displacement. During the early stages of operation, the displacement of the horizontal measurement point $DK_1$ essentially tends to be stable, and until the stable phase of operation, the increment of the horizontal displacement is only roughly 0.0001m. It is evident that the horizontal displacement is not significantly impacted by the rheology of the dam base. In Table 6, the settlement measuring point $SW_1$ in each working condition changes in the same time order as the measured value, and both of them increase with the passage of time. It is almost possible to ignore the error between the

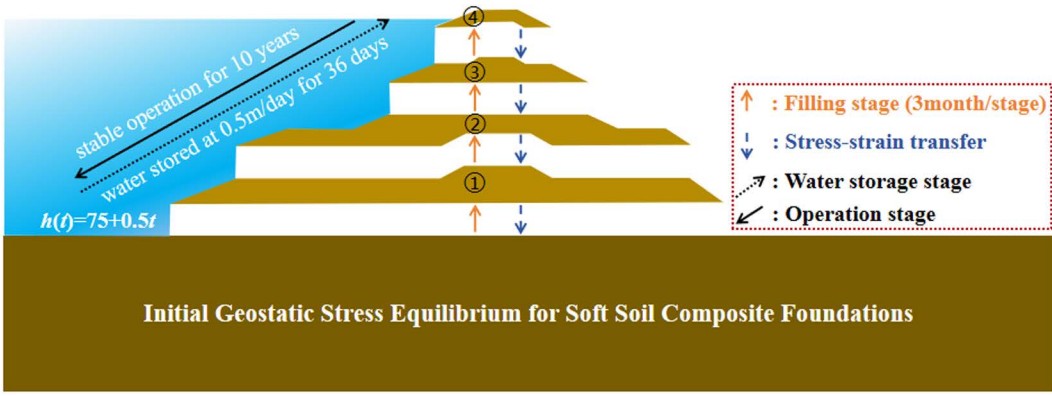

**Fig 9. Modeling flowchart.**

**Table 4. Schedule corresponding to each work period.**

| work status | timing |
| --- | --- |
| filling completion period | 12 months |
| initial impoundment period | 36 days |
| operational stabilization period | about 10 years |

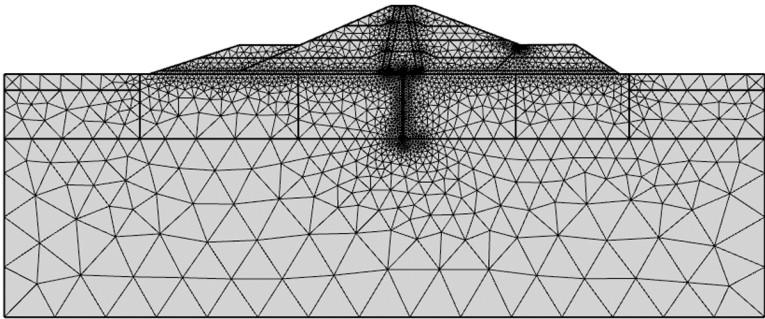

**Fig 10. Grid division diagram of the model.**

estimated and measured values, as indicated by the highest and minimum error rates of 2.14% and 0.47%, respectively. The observed values of settlement displacement and horizontal displacement are generally in agreement with the model's estimated results, indicating the correctness of the model's calculations and their compliance with the fundamental law of dam displacement.

**4.5.2. Verification of stress results.** Two monitoring points, $SG_1$ and $SG_2$, were chosen to get the calculated stress values of the measurement points, as illustrated in Fig 12. Under three working conditions based on the distribution law of stress calculation results and the dam body's stress monitoring position, Table 7 displays the comparison findings between the measured values and the model's computed values.

Table 7 shows that the computed values of the two models and the observed peak stress of the dam body during impounding are essentially in agreement, with the majority of the errors being less than 3%. The model in this paper matches the measured results better than the solidification model calculations, which show that after the operation is

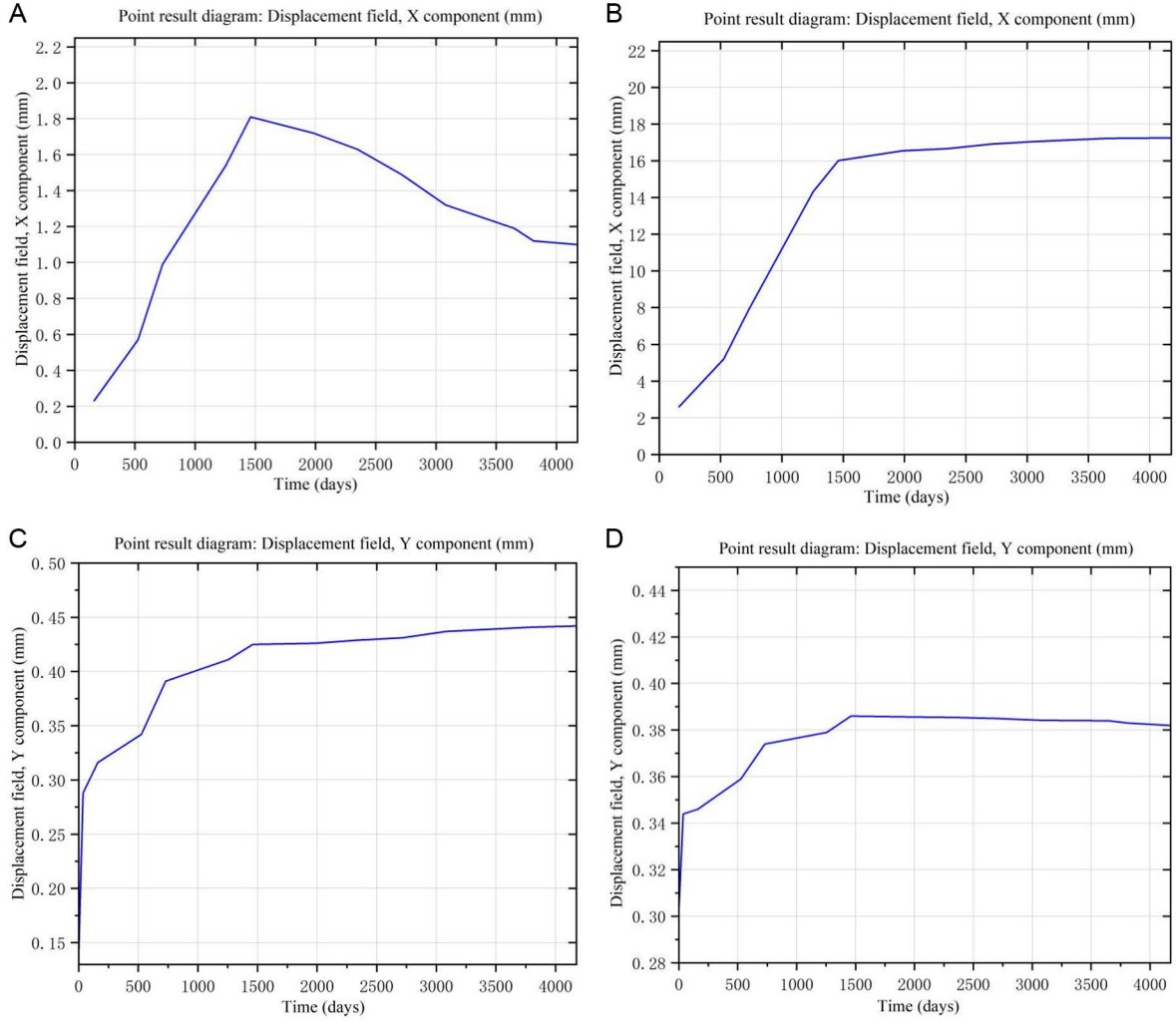

**Fig 11. Change curve of measuring point displacement with time. (a)** Point $DK_1$ result diagram. **(b)** Point $DK_2$ result diagram. **(c)** Point $SW_1$ result diagram. **(d)** Point $SW_2$ result diagram.

**Table 5. Comparison of measured and calculated horizontal displacements for different operation periods.**

| Measuring point | Running time | Horizontal displacement (mm) | | |
|---|---|---|---|---|
| | | Measured value | Calculated value | Errors |
| $DK_1$ | 2 years of operation | 0.9023 | 0.9953 | 9.34% |
| | 4years of operation | 1.81 | 1.81 | 0.00% |
| | 10 years of operation | 1.302 | 1.191 | 9.32% |
| $DK_2$ | 2 years of operation | 5.99 | 8 | 12.63% |
| | 4years of operation | 17 | 16.01 | 6.18% |
| | 10 years of operation | 19 | 17 | 9.40% |

**Table 6. Comparison of measured and calculated settlement values in different operating periods.**

| Measuring point | Running time | Settlement value (mm) | | |
|---|---|---|---|---|
| | | Measured value | Calculated value | Errors |
| SW$_1$ | filling completion | 0.137 | 0.14 | 2.14% |
| | water storage completed | 0.285 | 0.288 | 1.04% |
| | 2 years of operation | 0.393 | 0.391 | 0.51% |
| | 4 years of operation | 0.423 | 0.425 | 0.47% |
| | 10 years of operation | 0.437 | 0.44 | 0.68% |
| SW$_2$ | filling completion | 0.286 | 0.303 | 5.61% |
| | water storage completed | 0.325 | 0.344 | 5.52% |
| | 2 years of operation | 0.357 | 0.362 | 1.38% |
| | 4 years of operation | 0.394 | 0.386 | 2.07% |
| | 10 years of operation | 0.397 | 0.34 | 16.76% |

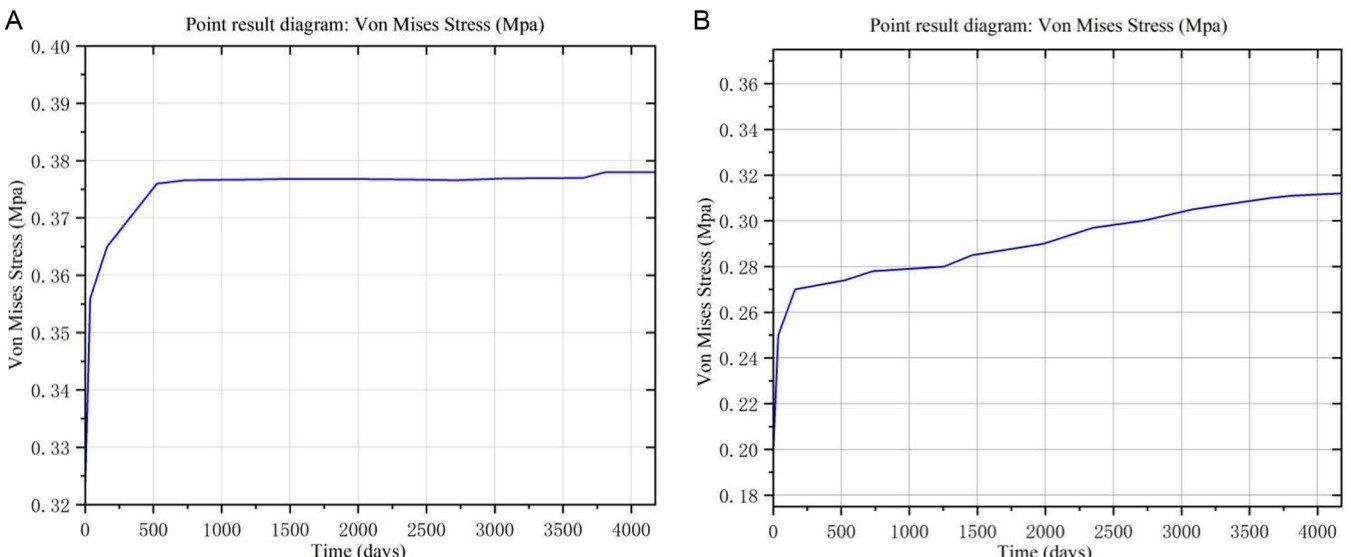

**Fig 12. Change curve of measuring point stress with time. (a) Point SG$_1$ result diagram. (b) Point SG$_2$ result diagram.**

stabilized, the model is small compared with the measured value, reduced by 0.004MPa and 0.02MPa, respectively, with a error rate of 1.05% and 6.45%. It can be seen that the proposed model is closer to the measured results. The data in Table 7 also shows that the rheology of the dam foundation has a relatively large influence on the stress of the dam body in the later period of operation.

**4.5.3. Dam seepage validation.** The comparison of the measured and calculated values of pore water pressure in the dam body at various operating times was obtained as indicated in Table 8 to confirm the accuracy of the seepage calculation results, along with the monitoring data of the seepage manometer in the dam body and the calculation results of the consolidation model.

Table 8 shows that the two models' calculated values and the pore pressure of the dam body both steadily drop as running time increases. With an increment of 2%–3%, the consolidation model's outcomes are greater than the measured values in the first four years of operation when compared to the measured values. Following four to five years of

**Table 7. Comparison of stresses at two measuring points within the dam body.**

| Measuring point | Running time | Stress(MPa) | | |
|---|---|---|---|---|
| | | Model value | Measured value | Errors |
| SG$_1$ | filling completion | 0.323 | 0.328 | 1.55% |
| | water storage completed | 0.356 | 0.36 | 1.12% |
| | stable operation | 0.377 | 0.381 | 1.05% |
| SG$_2$ | filling completion | 0.2 | 0.205 | 2.5% |
| | water storage completed | 0.25 | 0.243 | 2.8% |
| | stable operation | 0.31 | 0.33 | 6.45% |

**Table 8. Comparison of pore pressures in the clay core of dam.**

| running time/year | pore water pressure/MPa | | |
|---|---|---|---|
| | measured value | model | solidification model |
| 0.5 | 0.94 | 0.97 | 1.03 |
| 2 | 0.85 | 0.87 | 0.92 |
| 4 | 0.79 | 0.75 | 0.86 |
| 6 | 0.785 | 0.772 | 0.85 |
| 8 | 0.773 | 0.759 | 0.846 |
| 10 | 0.754 | 0.755 | 0.841 |

operation, the results progressively converge to a 0.85MPa steady value. It is evident that during the early phases of operation, the seepage field changes mostly due to the consolidation of the dam foundation, with rheology having a little effect. The computed results of pore water pressure agree with the measured values from the initial stage to the later stage. After four to five years of operation, the pore pressure gradually decreases in comparison to the consolidation model, suggesting that the rheological action of the dam foundation has some influence on the seepage field after consolidation is finished. The pore pressure decreases year by year and tends to be stable with the increase of time. It is obvious that the rheological effect of soft soil cannot be disregarded in the calculation, it speeds up the fluid-solid coupling process and validates the accuracy of the seepage calculation results. This is in accordance with the law of seepage field modification of the composite dam foundation.

## 5. Results analysis

Through the validation of the aforementioned model, the calculated results of deformation, stress, and seepage in the earth-rock dam are consistent with the engineering measurements, demonstrating that the model's computational accuracy can effectively reflect the influence of rheological effects on various key parameters. This section will systematically investigate the evolution of multi-physical characteristics of the composite foundation over time under three working conditions. Quantitative analyses will be conducted from the perspectives of deformation, stress, and seepage, respectively, to reveal the critical thresholds controlling the long-term stability of the foundation.

### 5.1. Analysis of dam deformation

**5.1.1. Horizontal displacement calculation analysis.** Fig 13 shows the cloud map of the calculation results of dam horizontal displacement under three working conditions.

The horizontal displacement cloud map shows that the maximum displacement in the upstream and downstream are 0.032 m and 0.036 m, respectively, and that the horizontal displacement at filling completion is essentially distributed

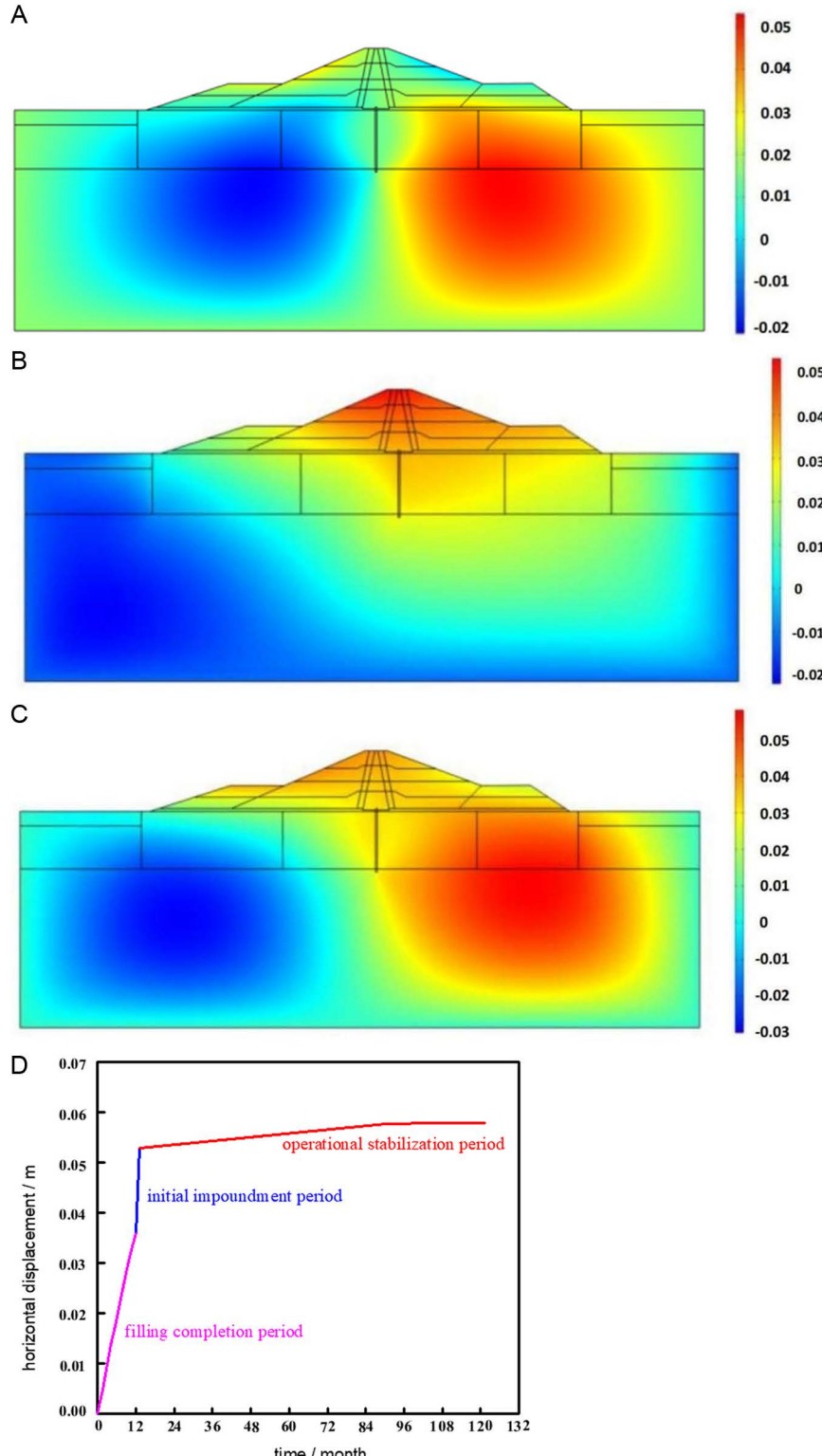

**Fig 13. Horizontal displacement distribution and maximum displacement change. (a)** Filling completion period. **(b)** Impoundment completion period. **(c)** Operational stabilization period. **(d)** Horizontal displacement with time process line.

symmetrically in both directions of the dam axis. The horizontal displacement in most parts of the dam body shifts downstream during the impoundment stage as the reservoir water level increases uniformly due to the action of water pressure, when impoundment period is finished, the highest value is 0.053m. Under the rheological and fluid-solid coupling effects, the horizontal displacement during operation tends to be stable with time, and is symmetrically distributed along the upstream side of the clay core, with a maximum value of 0.058m. The cumulative change process of maximum horizontal displacement at each stage is shown in Fig 13(d).

Fig 13(d) illustrates how the horizontal displacement of the dam body essentially changes linearly with time during the construction of the dam and the impoundment period. The impounding period's growth rate is significantly larger than the filling period's, indicating that impounding has a noticeable impact on horizontal displacement.Under the consolidation rheology and fluid-solid coupling impact of the dam foundation, the increase rate of horizontal displacement in the first four years of operation is faster than that in the last six years of operation during the stable operating phase following the completion of water storage. Nonetheless, the dam foundation's horizontal displacement only rises by 0.008 meters during the course of the operation, showing a comparatively sluggish growth tendency. It is evident that, following stable dam operation, the impact of the rheology of the dam foundation on the horizontal displacement is negligible.

**5.1.2. Calculation and analysis of settlement.** The cloud map of the dam settlement estimates for the three working circumstances are displayed in Fig 14.

As shown in Fig 14, the settlement of the dam after filling completed is essentially symmetrical along the dam axis. The area close to the clay core and cutoff wall experience the highest settlement, which is 0.54 meters. The upstream side of the dam body and foundation experiences the majority of settlement due to the influence of water weight after impounding completed, with the downstream side experiencing a clearly smaller change in settlement than the upstream side. Following an extended period of operation, the dam's settlement essentially becomes stable and distributes symmetrically along the dam's axis.

Fig 14(d) illustrates how the dam body's settlement essentially increases quickly over time during the filling and storage phases. The settlement amount following filling completion is 60.7% of the entire amount. Although the settlement is 0.13 meters in the water storage period, it is growing at a far faster rate than it did during the fill period, so the effects of water storage on the settlement are more noticeable. After impounding, the dam settlement will continue to increase with the consolidation rheology and fluid-solid coupling effect of the dam foundation, which will last for more than ten years, so its growth rate is obviously smaller than the first two stages, but compared with the impounding section, there is still an increment of 0.22m. It is evident that dam settlement is significantly impacted by the rheological effect of dam foundation consolidation.

## 5.2. Dam stress analysis

The results of the calculations demonstrate that, under the three working conditions, the distribution laws of large and small primary stresses are essentially the same. Fig 15 displays the distribution laws for each working condition, using the distribution results of large principal stresses as an example.

The big primary stress distribution graphic shows that the dam stress essentially distributes symmetrically along the dam axis when filling is complete, and that the stress clearly reduces as it passes through the cutoff wall and the clay core. This is because the impermeable material deforms differently than the foundation and dam shell materials. Following filling, the dam body experiences the "arch" effect, and the stress in the clay core clearly diminishes.Following the completion of water storage, the fuction of water pressure clearly increases the stress in the lower portion of the dam body and the dam foundation. Additionally, due to the material's varying deformation properties, the top and bottom of the concrete cutoff wall exhibit concentration of stress. The stress distribution of the dam body and foundation shifts downstream due to seepage pressure, and the stress beneath the foundation progressively rises.

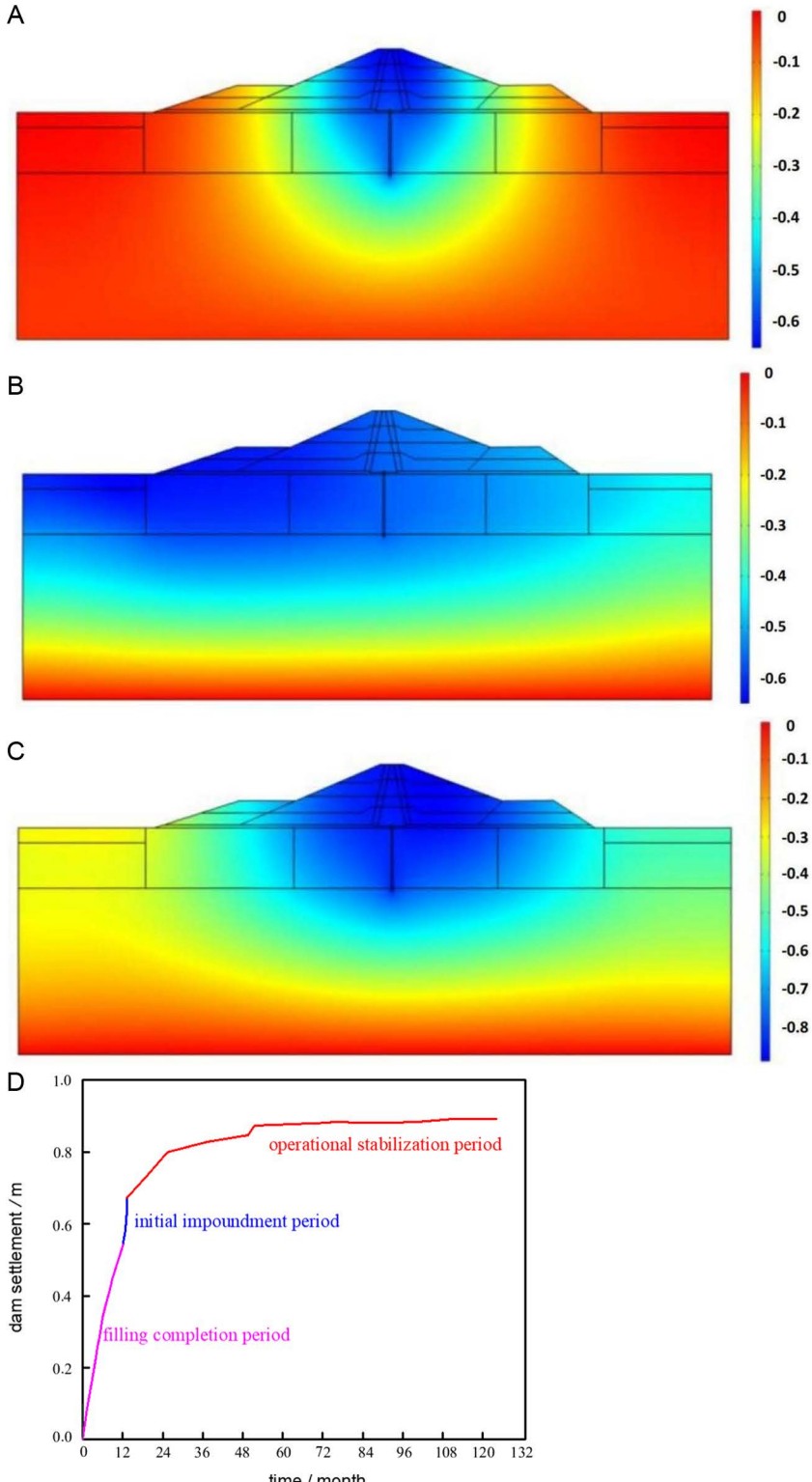

**Fig 14. Settlement distribution and maximum settlement change.** (a) Filling completion. (b) Water storage completed. (c) Operational stabilization period. (d) Settlement with time process line.

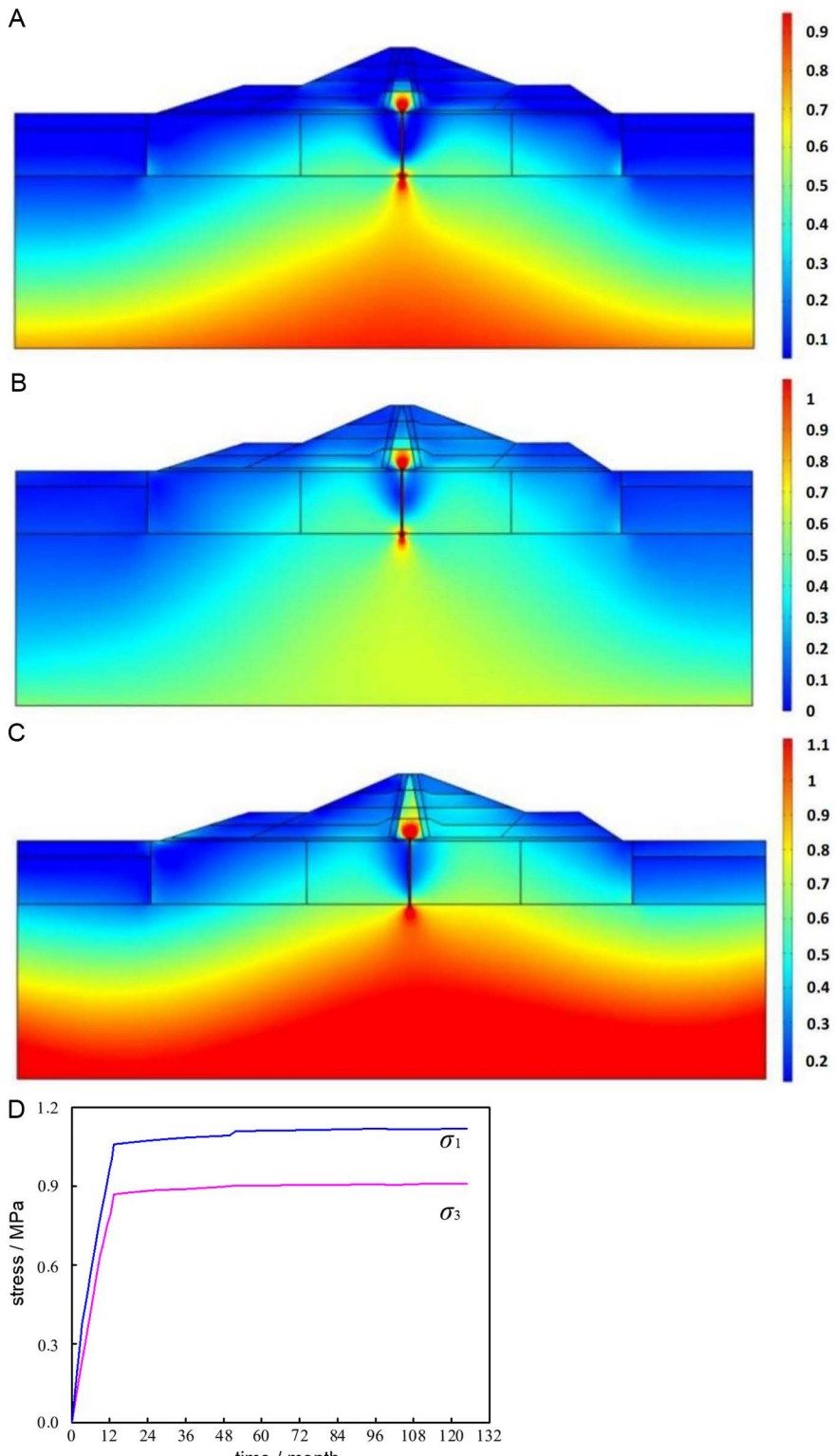

**Fig 15. Large principal stress distribution and stress peak change.** (a) Filling completion period. (b) Impoundment completion period. (c) Operational stabilization period. (d) Cumulative variation curve of major and minor principal stresses of dam body.

Peak stresses under the three operating conditions are 0.97MPa, 1.06MPa, and 1.12MPa, respectively, with a significant rise of 0.09MPa and 0.06MPa. Of them, the average monthly stress increment was 0.0075MPa during the impoundment, and the average monthly stress increment during the operation time was 0.0005MPa, or 1/15 of the average monthly stress increment during the water storage period. It is evident that the water load significantly affects the stress on the dam body as the water level rises during the impoundment stage. Fig 15(d) shows that the growth rate of peak stress of the dam body under various working conditions is as follows: filling period＞impoundment period＞operation period. During the operation period, the stress of the dam body continues to increase under the action of consolidation rheology and fluid-solid coupling. After 4 years of operation, the peak value of the dam body fluctuates little, but it continues to increase.After about 10 years of operation, the peak stress growth rate is basically unchanged, indicating that the overall stress state of the dam tends to be stable, the process lasts for a long time, and the growth rate gradually slows down.

### 5.3. Dam seepage analysis

Fig 16 shows the cloud map distribution of the seepage field of the dam after the completion of impoundment, after 1 year of operation and stable operation.

As can be seen from Fig 16, variations in the seepage field are mostly concentrated inside the clay core of dam as running time after water storage increases. The water blocking effect of the clay core and the cutoff wall is clearly seen when the seepage flow line flows downstream around them after the impoundment is complete. During the initial stage of operation, the seepage flow gradually developed downstream, the stress increased gradually, and the pressure head line inside the clay core progressively increased. The pressure head line in the clay core tends to remain steady during stable operation, and the pressure head clearly drops after passing through the clay core. The pressure water head line in the dam body bypasses the clay core and cutoff wall and percolates downstream along the dam shell and foundation after water storage is complete. This is primarily because the permeability coefficient of the materials used to make the dam shell and foundation is almost five orders of magnitude higher than that of the clay core and cutoff wall, respectively. The composite impervious system integrating the clay core and cutoff wall demonstrates effective seepage control in the dam structure, significantly reducing hydraulic permeation through the foundation. The pressure head curve in the clay core area gradually rises, the pressure head in the cutoff wall declines rapidly, and the pressure head of shell material on the top and lower reaches of the dam body is essentially horizontal. Further confirming the viability of the model's computation in this article is the running seepage law's agreement with the general seepage law in clay core rockfill dam.

## 6. Discussion

### 6.1. Importance of considering the rheology of the underlying soft soil

Engineering practice shows that the deformation (long-term deformation) of earth-rock dam will continue to increase with time after completion. According to current understanding, the long-term deformation of earth-rock dam may come from the rheological and humidification of composite materials of dam body and dam foundation, as well as the cumulative deformation caused by cyclic stress (load) fatigue and cyclic degradation of temperature and humidity, etc. These deformation may also have joint influence and superposition. The covering layer of deep soft soil in China is mostly flexible composite foundation, the upper part of which is reinforced by gravel pile or cement mixing pile, and the lower part is soft soil bearing layer.This kind of dam foundation will cause greater deformation of the dam body, and the rheological effect of the lower soft soil is ignored by the conventional calculation method, and the calculated results are inconsistent with the actual situation. The numerical results of this paper also reflect the persistence of the rheological effect under the composite foundation. With the passage of time, the stress, deformation and seepage tend to be stable after 4–5 years of operation of the consolidation model without considering rheology. The calculation results of this model show that the dam stress and deformation indexes continue to increase after 4–5 years of operation, although the increase is small,

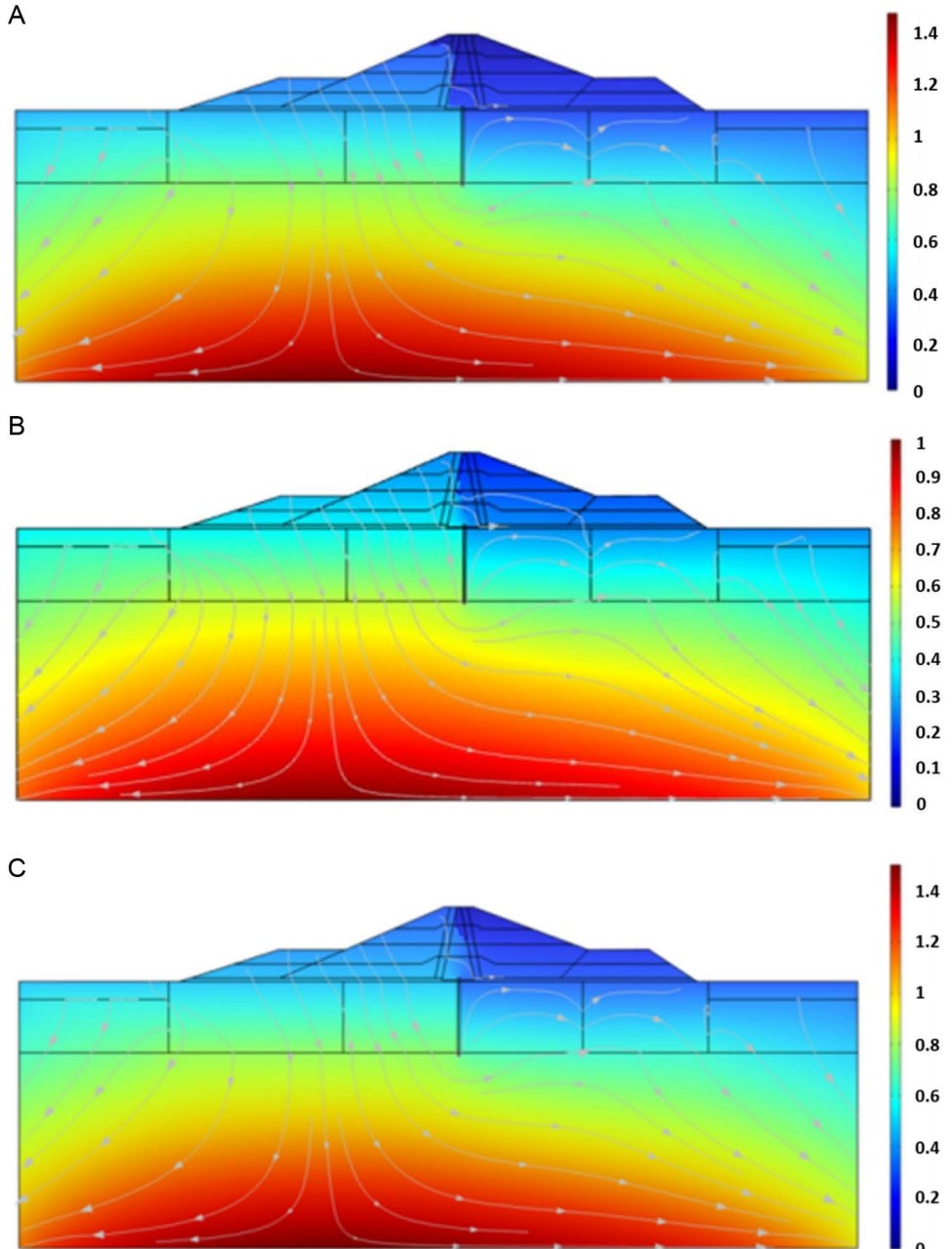

**Fig.16. Distribution of seepage field of dam at different stages (unit: MPa). (a)** Completion of impoundment. **(b)** Operation for 1 year. **(c)** Stable operation.

and the increase does not stop after more than 10 years of calculation. This indicates that the consolidation settlement of the composite foundation is basically completed within 4–5 years after operation, with a relatively large annual increase, during which time attention should be focused on the safety and stability of each structure of the dam. Thereafter, safety and stability problems occurring in the dam are basically caused by the rheological effect of the dam foundation.

The research results also reflect that the deformation, stress and permeability of the dam in the later period of operation will be affected by the underlying stratum rheological effect under the deep soft soil composite foundation. It can be seen that ignoring the rheology of soft soil will lead to the later settlement and deformation of earth-rock dam exceeding expectations, and the internal cracks or structural damage due to uneven settlement will affect the safety and stability of the dam body. It is relatively difficult to strengthen the dam after operation. If rheological effect is not considered in the design and treatment of dam foundation in the early stage, it will also leave security risks in the later stage. Therefore, when constructing a dam on a flexible composite foundation, it is crucial to scientifically address the rheological effects of soft soil and implement appropriate engineering measures. These measures may include extending the preloading period of the foundation before dam construction, installing a three-dimensional drainage system with longitudinal and transverse arrangements within the foundation, adopting intermittent staged loading during the dam construction phase, and increasing the penetration depth of piles in the reinforced zone. Such engineering controls ensure adequate consolidation of the soft soil, effective dissipation of pore water pressure, enhanced bearing capacity of the reinforced zone, and long-term continuous monitoring of dam settlement.

### 6.2. Rheological research tools for soft soil composite dam foundations

As mentioned in the introduction, domestic and foreign scholars have proposed different models according to different rheological time development laws or different stress influence mechanisms. In terms of research means, the side-limit compression rheometer is mostly used abroad, while the large triaxial rheometer is mostly used domestically. In order to study the fine-scale mechanism of rheological deformation, triaxial test with CT is also used to study the motion law of particles in rheology. Discrete elements are also used in the fine-scale analysis of rheology, and the time effect is modeled by the folding down of the interparticle bond strength with time or subcritical crack extension. In addition, previous studies on the long-term deformation characteristics of dam foundation composites were mainly limited to indoor rheological tests, and it is obviously inappropriate to analyze and predict the long-term deformation of high earth dams based on the results of such tests, and it is difficult to directly apply the rheological model established solely on the basis of the results of the indoor rheological tests in practical engineering. A practical and feasible method is to establish the corresponding rheological empirical model of complex materials on the basis of the analysis of the monitoring results and rules of the actual dam displacement, based on the measured deformation rules of the dam body, combined with the qualitative rules given by the laboratory rheological test. Only in this way, the empirical model containing various influencing factors can make a more reliable prediction of the long-term deformation of earth-rock dam. In this paper, the combination of parametric inversion and experimental validation to obtain the experimental parameters of the dam foundation material, the use of COMSOL multi-field coupled finite element to establish a new model and the comparison with the monitoring data to verify, is a relatively effective exploration method.

## 7. Conclusion

[1] The combined rheological model can more accurately depict the stress, deformation, and seepage conditions in each stages of the dam for the deep soft soil dam foundation strengthened with composite foundation.

[2] The combined rheological model's stress and deformation are 2.7% and 3% higher, respectively, than those of the consolidated model. It is important to pay attention to the increment since it will have an impact on the stability and safety of the dam's future operation.

[3] During the dam filling period, the total settlement accounts for more than 60%, but the increment per unit time is small. In the impoundment stage, due to the consolidation rheology and fluid-solid coupling of dam foundation, the strength and stiffness of each part of the dam are most affected, and the growth rate is also the largest. The unit time growth rates of stress and settlement are 0.00075MPa/ month and 0.108m/ month, respectively. During the operation period,

the growth rate of each index is relatively slow, but due to the longer operation time, its deformation and stress increment may lead to the instability or destruction of the main structure.

[4] The lower portion of the reservoir's naturally occurring soft soil composite foundation rheology is persistent. During normal reservoir operation, the composite foundation essentially completes consolidation settlement in 4–5 years, and the settlement is then altered by decades' worth of rheological influence.

[5] The models proposed in this paper for deep overlays are all based on transverse-isotropic media, which can not fully satisfy the actual engineering situation. For the study of rheological effect under the coupling of seepage stress, the rheological characteristics of anisotropic rock and soil should be considered in the future, so as to provide a theoretical basis for the construction of dams in the deep cover layer of soft soil in coastal, inland waterway or lake areas.

## Symbol list

| Symbol | Name | Unit |
|---|---|---|
| $\sigma$ | stress | Pa |
| $t$ | loading time | s |
| $E$ | elastic modulus | Pa |
| $\eta$ | viscosity coefficient | – |
| $\alpha$ | fractional order | – |
| $\varepsilon$ | strain | – |
| $\tau$ | shear stress | Pa |
| $D^{\alpha}$ | Riemann-Liouville fractional-order differential operator | – |
| $\Gamma()$ | Gamma function | – |
| $s$ | Laplace variable | – |
| $\bar{\sigma}$ | Laplace transforms of $\sigma$ | Pa |
| $\bar{\varepsilon}$ | Laplace transforms of $\varepsilon$ | – |
| $E_{a}()$ | single parameter Mittag-Leffer function | – |
| $\mu$ | poisson's ratio | – |
| $k$ | permeability coefficient | cm/s |
| $n$ | porosity | % |
| $v$ | seepage velocities | m/s |
| $\beta$ | oundation bearing capacity parameter | kPa |
| $\gamma$ | bulk density | N/m³ |
| $c$ | material cohesion | N/m² |
| $\varphi$ | angle of internal friction | ° |
| $K$ | modulus of elasticity | – |
| $n_{s}$ | elastic modulus index | – |
| $R_{f}$ | break ratio | – |
| $G,F,D$ | lateral deformation coefficient(E-umodel parameter) | – |
| $K_{ur}$ | coefficient of bulk modulus | – |
| $\rho$ | density | kg/m³ |
| $C_{m}$ | water capacity | – |
| $g$ | gravitational acceleration | m/s² |
| $S_{e}$ | saturation | – |
| $S$ | storage coefficient of porous media | m/s² |
| $p$ | pressure | Pa |
| $k_{s}$ | saturated permeability | md |

| Symbol | Name | Unit |
|---|---|---|
| $k_r$ | relative permeability | – |
| $H_t$ | total head | m |
| $H$ | position head | m |
| $Q_m$ | fluid source-sink term | m³/h |
| $\nabla$ | gradient operator | – |
| $S_{ext}$ | saturation coefficient | – |
| $P_f$ | pore pressure | Pa |
| $p_{ref}$ | reference pressure level | Pa |
| $I$ | | |
| $\alpha_B$ | Bio-consolidation parameter | – |
| $\varepsilon_{vol}$ | porous medium body strain | – |
| $\boldsymbol{\sigma}$ | Cauchy stress tensor | Pa |
| $\boldsymbol{\varepsilon}$ | strain tensor | – |
| $\boldsymbol{C}$ | elasticity matrix | Pa |
| $I$ | unit matrix | – |
| $M$ | Biot modulus | Pa |
| $\zeta$ | value of the variation of the fluid content in the porous medium | – |
| $P_m$ | volume change of soil particles | Pa |
| $K_s$ | elastic modulus of solid | Pa |

## Supporting information

**S1 Table. This is the horizontal displacement with time process line in Fig 13(d).**
(XLSX)

**S2 Table. This is the settlement with time process line in Fig 14(d).**
(XLSX)

**S3 Table. This is the cumulative variation curve of major and minor principal stresses of dam body in Fig 15(d).**
(XLSX)

## Author contributions

**Conceptualization:** Jin Qu.

**Data curation:** Jin Qu, Fan Wang, Xiaoju Wang.

**Formal analysis:** Haitao MAO.

**Funding acquisition:** Jin Qu, Haitao MAO, Xiaoju Wang.

**Investigation:** Sheng Tang.

**Methodology:** Jin Qu, Haitao MAO.

**Resources:** Haitao MAO.

**Software:** Jin Qu, Haitao MAO, Fan Wang.

**Supervision:** Haitao MAO.

**Validation:** Jin Qu.

**Writing – original draft:** Jin Qu.

**Writing – review & editing:** Jin Qu, Haitao MAO, Sheng Tang.

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
