## [Decision Letter · Decision Letter 0]

PONE-D-24-13121Research on composite foundation rheology using the combined rheological element modelPLOS ONE

Dear Dr. MAO,

Thank you for submitting your manuscript to PLOS ONE. After careful consideration, we feel that it has merit but does not fully meet PLOS ONE’s publication criteria as it currently stands. Therefore, we invite you to submit a revised version of the manuscript that addresses the points raised during the review process.

We look forward to receiving your revised manuscript.

Kind regards,

Ahmed M. Saqr, Ph.D.

Academic Editor

PLOS ONE

Additional Editor Comments:

Below are the comments of two reviewers regarding your submission to PLOS One. The two reviewers have made substantive critical comments, and you should pay close attention to them when making your revisions. The reviewers' comments are important, as they will assist you in making your paper much more interesting to our readers. Please, address properly all the comments of the reviewers. Please, submit your revised manuscript online by using the Editorial Manager system.

Reviewers' comments:

Reviewer's Responses to Questions

**Comments to the Author**

1. Is the manuscript technically sound, and do the data support the conclusions?

Reviewer #1: Yes

Reviewer #2: Yes

2. Has the statistical analysis been performed appropriately and rigorously? 

Reviewer #1: Yes

Reviewer #2: Yes

3. Have the authors made all data underlying the findings in their manuscript fully available?

Reviewer #1: Yes

Reviewer #2: Yes

4. Is the manuscript presented in an intelligible fashion and written in standard English?

Reviewer #1: Yes

Reviewer #2: Yes

5. Review Comments to the Author

Reviewer #1: The overall structure of the paper is relatively sound, the results are abundant, and the conclusions are credible. However, the following issues still exist:

1. The experimental parameters derived from Table 1, along with the basic information and results of the experiments, need to be provided.

2. The specific layout diagram of monitoring points needs to be incorporated.

3. It is necessary to slightly adjust the structure of the paper. Firstly, validate the rationality of this model, for example, by illustrating the differences between calculated displacement values and monitoring values. Utilizing a histogram might not provide the most intuitive representation; I suggest employing alternative methods. Following this, discuss the relevant patterns of change.

4. Does the consolidation theory used here specifically refer to the Biot consolidation theory, or something else?

Reviewer #2: Thank you for inviting me to review the paper entitled “Research on composite foundation rheology using the combined rheological element model”. The topic of the article is interesting. I recommend reconsideration of the paper after addressing the following modifications:

• What is the novelty of this manuscript? It should be clearly mentioned in the last paragraph of the introduction.

• Please, replace all the figures with high-resolution ones.

• You should add the limitation of your study at the end of the paper before the conclusion section.

• What is your recommendation for future studies to enhance the applied methodology? You can discuss how to relate your research findings with sustainable development goals (SDGs) to achieve more environmental, economic, and social benefits. References to cite:

https://doi.org/10.1016/j.ejrh.2024.101703

https://doi.org/10.1016/j.gsd.2024.101087

https://doi.org/10.1007/978-981-99-4101-8_27

https://doi.org/10.1007/978-981-99-1381-7_6

6. PLOS authors have the option to publish the peer review history of their article (what does this mean?). If published, this will include your full peer review and any attached files.

Reviewer #1: No

Reviewer #2: No

---

## [Author Response · Author response to Decision Letter 1]

4 Jul 2024

Reviewer #1:

1.The experimental parameters derived from Table 1, along with the basic information and results of the experiments, need to be provided.

Response:Due to the large number of calculation parameters used in this paper, combined with the concrete engineering characteristics of the composite foundation of Dakai clay core rockfill dam, the final parameters are mainly obtained through empirical values, indoor and outdoor tests and inverse calculation. Firstly, Poisson's ratio μ and foundation bearing capacity parameter β are related to soil type, compactness, water content, friction Angle and other factors, and cannot be directly obtained through experiments. Therefore, empirical values recommended in literature [26] are adopted. The permeability coefficient k, bulk density γ and porosity v were obtained by outdoor sampling combined with indoor tests. Secondly, by using the mathematical formula of the improved rheological model and combining with the settlement monitoring data of the dam body during the filling period, the initial elastic modulus E and E1, the elastic modulus E0 and E2 of the creep model and the viscosity coefficient η of the creep model are obtained through inversion calculation. In order to verify the rationality of the creep parameter values of the reinforced area and the underlying layer, To ensure the accuracy of the numerical model calculation results, the inversion calculation parameters were compared with the rheological experimental parameters of soil and stone in each layer, as shown in Table 3. The results show that the model parameters determined by inversion calculation are feasible. For details, see lines 295-305 and 316-325 of the manuscript.

2.The specific layout diagram of monitoring points needs to be incorporated.

Response:We have added the monitoring system layout in the revised draft uploaded, as shown in Fig 6.

3.It is necessary to slightly adjust the structure of the paper. Firstly, validate the rationality of this model, for example, by illustrating the differences between calculated displacement values and monitoring values. Utilizing a histogram might not provide the most intuitive representation; I suggest employing alternative methods. Following this, discuss the relevant patterns of change.

Response:As you suggested, we have put the content of model validation after the modeling and before the analysis of results . After the establishment of the model, through the verification of displacement, stress and seepage, the measured values are basically consistent with the calculated results of the model, which proves the effectiveness of the model. Then the dam displacement, stress and seepage analysis are carried out.This adjustment makes the article more logical.In addition, for the problem that the histogram expression in the model verification content is not intuitive, we have listed the comparison results between the model and the measured values in the form of a table, and attached the error value between the two, so that the results can be clearly presented.

4.Does the consolidation theory used here specifically refer to the Biot consolidation theory, or something else?

Response:Yes, the consolidation theory used in the paper is Biot consolidation theory, highlighted in line 342 of the manuscript.

Reviewer #2:

1.What is the novelty of this manuscript? It should be clearly mentioned in the last paragraph of the introduction.

Response:The novelty of the manuscript is that the coupling model of seepage and stress under the consolidation rheological effect of the dam foundation is established in view of the current situation that the influence of the rheological effect of the soft soil composite foundation on the clay core rockfill dam is still unclear, which makes up for the deficiency of previous studies that only consider the consolidation and settlement behavior of the foundation while ignoring the rheological effect. This paper provides a theoretical basis for the study of rheological model of soft soil composite foundation with deep cover layer, and takes the Dakai clay core rockfill dam project on soft soil composite foundation as an example to define the influence mechanism of rheological foundation on clay core rockfill dam on soft soil composite foundation.We have revised the last paragraph of the introduction in the revised manuscript uploaded, in lines 99-117 .

2.Please, replace all the figures with high-resolution ones.

Response:According to your suggestion, I have replaced all the pictures in the manuscript with high-resolution ones, see the revised draft uploaded for details.

3.You should add the limitation of your study at the end of the paper before the conclusion section.

Response:The composition of each layer of geotechnical materials in the foundation with deep cover layer is complex, and the non-uniformity of each layer of materials inside the foundation will make the geotechnical characteristics of obvious anisotropy. However, considering the solvable and computational accuracy of the model, the models proposed in this paper for the deep cover layer are all built on the transverse isotropic medium, which does not fully meet the love situation of the ground in the actual project. In the future study, the rheological properties of anisotropic rock and soil, especially the rheological effects under the coupling of seepage stress, still need to be further analyzed and studied.We have supplemented this part in lines 651-656 of the manuscript.

4.What is your recommendation for future studies to enhance the applied methodology? You can discuss how to relate your research findings with sustainable development goals (SDGs) to achieve more environmental, economic, and social benefits. References to cite:

https://doi.org/10.1016/j.ejrh.2024.101703

https://doi.org/10.1016/j.gsd.2024.101087

https://doi.org/10.1007/978-981-99-4101-8_27

https://doi.org/10.1007/978-981-99-1381-7_6

Response:In the future research, for the simulation of composite soft soil foundation, the influence of factors such as the thickness and location of each layer of soil on the upper and lower layers of soil can be considered, and these effects can be parameterized and introduced into the analysis and calculation of the model. In addition, the complex environment of actual engineering can also be considered. Rheological properties of composite soft soil foundation under multi-field coupling conditions, such as temperature, stress, seepage, chemical reaction, etc., are of guiding significance for dam construction in the deep cover layer of soft soil in coastal, inland waterway or lake area.

---

## [Decision Letter · Decision Letter 1]

PONE-D-24-13121R1Research on composite foundation rheology using the combined rheological element modelPLOS ONE

Dear Dr. MAO,

Thank you for submitting your manuscript to PLOS ONE. After careful consideration, we feel that it has merit but does not fully meet PLOS ONE’s publication criteria as it currently stands. Therefore, we invite you to submit a revised version of the manuscript that addresses the points raised during the review process.

We look forward to receiving your revised manuscript.

Kind regards,

Qiang Li

Academic Editor

PLOS ONE

Reviewers' comments:

Reviewer's Responses to Questions

**Comments to the Author**

1. If the authors have adequately addressed your comments raised in a previous round of review and you feel that this manuscript is now acceptable for publication, you may indicate that here to bypass the “Comments to the Author” section, enter your conflict of interest statement in the “Confidential to Editor” section, and submit your "Accept" recommendation.

Reviewer #1: (No Response)

Reviewer #2: All comments have been addressed

Reviewer #3: (No Response)

2. Is the manuscript technically sound, and do the data support the conclusions?

Reviewer #1: (No Response)

Reviewer #2: Yes

Reviewer #3: Partly

3. Has the statistical analysis been performed appropriately and rigorously? 

Reviewer #1: (No Response)

Reviewer #2: Yes

Reviewer #3: Yes

4. Have the authors made all data underlying the findings in their manuscript fully available?

Reviewer #1: (No Response)

Reviewer #2: Yes

Reviewer #3: Yes

5. Is the manuscript presented in an intelligible fashion and written in standard English?

Reviewer #1: (No Response)

Reviewer #2: Yes

Reviewer #3: Yes

6. Review Comments to the Author

Reviewer #1: (No Response)

Reviewer #2: I see the manuscript, entitled: "Research on composite foundation rheology using the combined rheological element model" is now improved. The authors have addressed all comments. Accept!

Reviewer #3: The workload of the paper feels heavy and the innovation is not very obvious. The logical framework of the writing is not clear enough. Here are some specific suggestions:

1.The high number of parameters employed (see Table 1) requires a "list of symbols" section where the readers can easily access all the definitions.

2.To improve the readability and logic of this text, please precede all titles of the manuscript with serial numbers, such as "1, 2, 1.1, 2.1...".

3.The theoretical formulas (1)-(11) of the numerical model need to be given in the manuscript with corresponding references.

4.Add a meshing diagram of the dam model in the manuscript in COMSOL Multiphysics.

5.The title "Modeling" on line 350 is necessary to specify which model it refers to, the existing title makes the logical framework of the article very confusing. The narrative in this section may be described in the form of a flowchart.

6.For the model validation part, it is recommended to provide a 2D result diagram in COMSOL Multiphysics in the manuscript, which can more intuitively reflect the accuracy of the model.

7.It is necessary to improve the resolution of the picture, such as Figure 11, etc., the display of dense streamlines is not clear. It is recommended that the results in the text be represented by color cloud plots, discarding the contour plots. Simple contour plots are not conducive to analyzing the results.

8.“Comsol” should be “COMSOL”

9.Please check and revise the text for language problems.

7. PLOS authors have the option to publish the peer review history of their article (what does this mean?). If published, this will include your full peer review and any attached files.

Reviewer #1: No

Reviewer #2: No

Reviewer #3: No

---

## [Author Response · Author response to Decision Letter 2]

25 Oct 2024

Replies to the reviewers’ comments:

Reviewer #3:

1.The high number of parameters employed (see Table 1) requires a "list of symbols" section where the readers can easily access all the definitions.

Response:According to your suggestion, we have sorted out all the parameter symbols in the manuscript, and modified some places (for example, the same symbols were used to express different parameters), and the corresponding highlights were also made in the manuscript. Our list explains the meaning of each parameter symbol, along with the corresponding units, and finally places the symbol list after Conclusion, specifically at line 556 in the revised manuscript.

2.To improve the readability and logic of this text, please precede all titles of the manuscript with serial numbers, such as "1, 2, 1.1, 2.1...".

Response:According to your suggestion, we have numbered the main body of the manuscript. The first, second and third headings are denoted by 1, 1.1 and 1.1.1 respectively. The specific changes have been highlighted in yellow in the revised manuscript.

3.The theoretical formulas (1)-(11) of the numerical model need to be given in the manuscript with corresponding references.

Response:According to your suggestion, we have added references to formulas (1) - (4), specifically see references [24] and [27-29]. Formula (4) is summarized according to the research ideas in the literature rather than directly citing the formula in the literature [29], and mainly expresses the total stress and total strain of the fractional merchant model.In addition, on the basis of formula (4), formulas (5) - (10) are derived by Laplace transform and inverse transformation, and formula (11) is the boundary condition given to the rheological model of composite foundation, without citing relevant literature. At the same time, thank you very much for asking this question, because in the process of replying to this question, we found some writing errors in the formula and corrected them in time, please see the revised manuscript for details.

4.Add a meshing diagram of the dam model in the manuscript in COMSOL Multiphysics.

Response:According to your suggestion, we have added the grid diagram of the dam model in the revised manuscript, as shown in Figure 8 of lines 322-323.

5.The title "Modeling" on line 350 is necessary to specify which model it refers to, the existing title makes the logical framework of the article very confusing. The narrative in this section may be described in the form of a flowchart.

Response:According to your suggestion, we deleted the titles 1, 2, and 3 in the modeling section, and adjusted the text expression of these parts to make the structure consistent with the full text. At the same time, the flow chart generated according to the model building steps is also added to show the model generation process more clearly. See Figure 7 on lines 322-323 for details.

6.For the model validation part, it is recommended to provide a 2D result diagram in COMSOL Multiphysics in the manuscript, which can more intuitively reflect the accuracy of the model.

Response:According to your suggestion, we have retrieved the point result graphs of measuring points DK1, DK2, SW1, SW2, SG1 and SG2 in COMSOL, so as to intuitively reflect the correctness of the model. For details, see Fig 9 and Fig 10 in the revised manuscript.

7.It is necessary to improve the resolution of the picture, such as Figure 11, etc., the display of dense streamlines is not clear. It is recommended that the results in the text be represented by color cloud plots, discarding the contour plots. Simple contour plots are not conducive to analyzing the results.

Response:According to your suggestion, we have changed all the contour maps of horizontal displacement, settlement, stress and seepage field in the manuscript into color cloud maps. The analysis of the results in the original manuscript was completed on the basis of contour maps. After changing to cloud maps, we re-analyzed the results of the cloud maps, so the text expression of this part is somewhat different from the original one. See line 400-419, 428-443, 455-477 and 484-500 for details.

8.“Comsol” should be “COMSOL”

Response: We checked the manuscript for typos regarding Comsol and changed all Comsol to COMSOL, see lines 11, 80, 296, and 331 for details.

9.Please check and revise the text for language problems.

Response:According to your suggestions, we have carefully checked the language problems in the manuscript and made revisions, which have been highlighted in green in the revised manuscript. At the same time, if you feel that the manual correction has not met the requirements of the paper publication, we can further polish the manuscript in the later stage.

---

## [Decision Letter · Decision Letter 2]

PONE-D-24-13121R2Research on composite foundation rheology using the combined rheological element modelPLOS ONE

Dear Dr. MAO,

Thank you for submitting your manuscript to PLOS ONE. After careful consideration, we feel that it has merit but does not fully meet PLOS ONE’s publication criteria as it currently stands. Therefore, we invite you to submit a revised version of the manuscript that addresses the points raised during the review process.

Please submit your revised manuscript by Jan 12 2025 11:59PM. If you will need more time than this to complete your revisions, please reply to this message or contact the journal office at plosone@plos.org. Please include the following items when submitting your revised manuscript:

We look forward to receiving your revised manuscript.

Kind regards,

Huaming An

Academic Editor

PLOS ONE

Journal Requirements:

Reviewers' comments:

Reviewer's Responses to Questions

**Comments to the Author**

1. If the authors have adequately addressed your comments raised in a previous round of review and you feel that this manuscript is now acceptable for publication, you may indicate that here to bypass the “Comments to the Author” section, enter your conflict of interest statement in the “Confidential to Editor” section, and submit your "Accept" recommendation.

Reviewer #1: All comments have been addressed

Reviewer #3: (No Response)

Reviewer #4: (No Response)

Reviewer #5: All comments have been addressed

Reviewer #6: All comments have been addressed

Reviewer #7: (No Response)

Reviewer #8: All comments have been addressed

2. Is the manuscript technically sound, and do the data support the conclusions?

Reviewer #1: (No Response)

Reviewer #3: No

Reviewer #4: (No Response)

Reviewer #5: Yes

Reviewer #6: Yes

Reviewer #7: Partly

Reviewer #8: Yes

3. Has the statistical analysis been performed appropriately and rigorously? 

Reviewer #1: (No Response)

Reviewer #3: No

Reviewer #4: (No Response)

Reviewer #5: Yes

Reviewer #6: Yes

Reviewer #7: Yes

Reviewer #8: Yes

4. Have the authors made all data underlying the findings in their manuscript fully available?

Reviewer #1: (No Response)

Reviewer #3: (No Response)

Reviewer #4: (No Response)

Reviewer #5: Yes

Reviewer #6: Yes

Reviewer #7: No

Reviewer #8: Yes

5. Is the manuscript presented in an intelligible fashion and written in standard English?

Reviewer #1: (No Response)

Reviewer #3: No

Reviewer #4: (No Response)

Reviewer #5: Yes

Reviewer #6: Yes

Reviewer #7: No

Reviewer #8: Yes

6. Review Comments to the Author

Reviewer #1: (No Response)

Reviewer #3: Although this manuscript has undergone extensive revisions by the authors, the issues within it still raise significant concerns for me. In fact, the depth of the article is far from adequate. Many expressions and figures are also not presented according to standards. After careful consideration, I firmly believe that this manuscript is not suitable for publication. The organization of the article is indeed lacking. The research methods, particularly the mathematical modeling section, appear quite chaotic and lack clarity. Additionally, the validation part is also questionable; it is unclear whether the monitoring data, such as displacement, was obtained accurately, and the manuscript does not provide detailed explanations, merely comparing the data with simulation results. The results and discussion section is also insufficiently thorough.

Reviewer #4: (No Response)

Reviewer #5: The quality of this manuscript has been relatively high after two rounds of revisions. And it is to be recognised and appreciated that the innovativeness of this manuscript is very clear, as the authors make it clear in the introduction that existing studies have ignored the effects of rheology. This study proposes a new combined rheological cell model based on the classical combined cell model and also simulates the seepage and displacement of a deep soft soil composite dam foundation using COMSOL software. However I think this manuscript should also address the following minor issues before acceptance:

1. It is suggested that the formula of 122 lines can be numbered.

2. On line 151, 'see in the picture' should be 'see in Fig.3'

3. The "Kpa" in the text should be "kPa", for example, line 187, line 190.

4. The author is requested to add more detailed information of the project example in "3.1 Summary of project", such as the location of the dam in China, etc. A special picture needs to be drawn, which should show a complete map of China and the location or coordinates of the project.

5. In "3.4 Monitoring system layout" of the manuscript, the author shows various kinds of sensors that can be used to monitor the displacement and seepage of DAMS. I am very curious about whether the author installed these sensors 11 years ago, dug up the dam and buried the sensors in it, or how they were installed inside the rock and soil body. It is well known that it is difficult to bury sensors into the interior of a geotechnical body, but the authors seem to have done it easily and buried many different types of sensors. And I'm also wondering, did the author just put the computer and collector directly on the site for 11 years, which doesn't seem realistic, or did he utilise some kind of wireless networked data transmission device to monitor the site remotely? None of this key information and details are mentioned by the authors in the latest version of the manuscript. The author does not describe in detail when the monitoring time started and ended, and the text does not contain any pictures of on-site installation of sensors and equipment. I don't think it's appropriate.

6. According to the description in this paper, these devices are quite advanced, the longest monitoring time can reach 4000 days, the equipment and sensors run continuously for more than 10 years, and the data acquisition instruments are constantly powered up during operation, which is very scary, and I am a little curious about how the author did it. In order to let readers have a better understanding of the equipment, it is best to add a special section detailing the content of the equipment and devices, introducing the models of the equipment and sensors used in the field test of this study, and it is best to put the schematic diagram of the equipment.

Reviewer #6: (No Response)

Reviewer #7: This paper establishes a new rheological model of soft soil composite foundations to study the deformation, stress and rheological characteristics of core wall rockfill dam and composite foundations. This topic is interesting and important for engineering practice. However, there are some outstanding issues that must be noted:

1. The papers writing logic is poor and fails to grasp the core description. Keywords are not accurate enough. For example, why is “elemental modeling” used as a keyword? It is suggested that the author should strengthen the revision of paper writing.

2. The description of “Soft soil composite foundation rheological modeling” in Part 2 is confused. For example, E2 in formula 4 and S in formula 6 are not explained, and Eα in the later explanation of formula 7 does not appear in the formula, so I do not know what it means.

3. Many technical terms in Part 3 are used incorrectly, such as clay core wall rockfill dam instead of clay heart wall rockfill dam, and the description of elevation should be used instead of height, unit kPa. Instead of Kpa;

4. The horizontal displacement, settlement and stress in Fig.7(b), Fig.8(b) and Fig.10(b) do not conform to the distribution law of deformation and stress during the storage period. It is recommended to check it carefully and modify it

Reviewer #8: The authors have made explicit changes and improvements to the paper in response to the reviewers' comments, as detailed below:

1. The authors have provided detailed maps and explanations of the experimental parameters and monitoring point layouts in the revised manuscript, supplemented by high-resolution relevant diagrams.

2. The authors clearly state the innovation of the paper in the introduction section.

3. The authors have optimized the structure of the paper, added a list of symbols and adjusted the title numbering to improve the logic and readability of the paper.

4. The authors have reorganized the content position of the model validation section to enhance the logic of the paper; and added a 2D COMSOL result chart and a model construction flow chart to further clarify the research process.

5. The authors have added a new section on research limitations, pointing out that the existing models do not sufficiently take into account anisotropy and multi-field coupling effects in practical applications, and suggesting directions for future research.

6. The authors have touched up the language of the article and fixes the terminology problem.

The revision of the article responds to the comments made by the reviewers, resulting in a significantly improved paper in terms of structure, content and scientific quality, which meets the requirements for publication, and a small amount of proofreading is recommended.

7. PLOS authors have the option to publish the peer review history of their article (what does this mean?). If published, this will include your full peer review and any attached files.

Reviewer #1: No

Reviewer #3: No

Reviewer #4: No

Reviewer #5: No

Reviewer #6: No

Reviewer #7: No

Reviewer #8: No

---

## [Author Response · Author response to Decision Letter 3]

11 Jan 2025

Dear academic editor and reviewers:

Thank you for your letter and the reviews’ comments on our manuscript entitled “Research on composite foundation rheology using the combined rheological element model ” (PONE-D-24-13121).Those comments are very helpful for revising and improving our paper,as well as the important guiding significance to other research.We have studied the comments carefully and made corrections which we hope meet with approval.The main corrections are in the revised manuscript and the responds to the reviews’ comments are as follows.

Replies to the reviewers’ comments:

Reviewer #3:

Although this manuscript has undergone extensive revisions by the authors, the issues within it still raise significant concerns for me. In fact, the depth of the article is far from adequate. Many expressions and figures are also not presented according to standards. After careful consideration, I firmly believe that this manuscript is not suitable for publication. The organization of the article is indeed lacking. The research methods, particularly the mathematical modeling section, appear quite chaotic and lack clarity. Additionally, the validation part is also questionable; it is unclear whether the monitoring data, such as displacement, was obtained accurately, and the manuscript does not provide detailed explanations, merely comparing the data with simulation results. The results and discussion section is also insufficiently thorough.

Response: First of all, I would like to express my sincere appreciation for the time and effort you have invested in reviewing our manuscript. We value the valuable comments you have provided and have carefully considered each piece of feedback.

1.Regarding the problem that the depth of the article is far from enough: In previous studies on the calculation of composite foundation models, only the consolidation and settlement behavior of the foundation is usually considered, while the rheological effect has a very significant influence on the stability and settlement of the foundation. Therefore, on the basis of the classical combined model of composite foundation, we established a coupled model of seepage and stress under the consolidation rheological effect of soft soil composite dam foundation, and took the Dakai clay core wall rockfill dam project on soft soil composite foundation as an example, established the calculation model and compared the observed and simulated values with the actual observation data. It is found that the difference between the established model and the observation results is not significant, which verifies the accuracy of the established model, so we believe that this manuscript provides a certain theoretical basis for the study of the rheological modeling of soft soil composite foundations with deep cover.

2.Regarding the problem of non-standard expressions and graphs: In the first revision, we realized the problem of non-standard mathematical expressions in the manuscript, so in the last round of revision, we reviewed the formulae for writing and labeling errors, and added a list of symbols after the conclusions to explain the meanings of all symbols appearing in the formulae of the manuscript, so that the readers can understand the meanings expressed by the formulae more clearly. As for the issue of substandard graphs, we apologize that we are not quite sure exactly what aspect you are referring to. Figures 1-3 are the descriptions of the model, and Figures 4-8 are the project location map, dam filling materials, dam foundation treatment measures, dam monitoring system layout schematic and dam monitoring data acquisition flow chart, respectively. According to your suggestions, we have further adjusted the clarity of the pictures, please see the revised manuscript with track changes for details. In the manuscript, the change diagrams (i.e., Fig.10-Fig.16) about the grid division, calculation cloud diagrams, measurement point displacement, measurement point stress, etc. are all the calculation cloud diagrams exported from COMSOL software, and we think that there is no problem with these diagrams in terms of clarity and graphical standardization.

3.Regarding the lack of clarity in the mathematical modeling section: According to your suggestion, we have reorganized the structure of the article, focusing on the mathematical modeling section to ensure that there is relevance and logic between each part. The specific changes are as follows: the original Chapter 3 (Analysis of engineering examples) is split into two chapters, Chapter 3 (Engineering overview) and Chapter 4 (Numerical modeling). The modified Chapter 3 mainly introduces the basic situation of Dakai Reservoir dam and the arrangement of the dam body monitoring system, while Chapter 4 mainly introduces the process of mathematical modeling, including model selection, control equations, model parameters, and validation of the model after modeling. This change is more in line with the steps of numerical model building and makes the article more clearly organized. Please see revised manuscript with track changes for details.

4. Regarding the validation part and the authenticity of the monitoring data: First of all, we understand your concern about the authenticity of the data, and we only provide the monitoring data at a specific time in the manuscript for two main reasons. One is that the soft soil dam base rheology is time-sensitive, the dam base rheology will evolve over time, and the effect on the dam base rheology is significant when the external action on the dam base changes, so this paper chooses several key periods to compare the simulated and calculated values of the model, such as the period of filling completion, initial impoundment, the first 2-4 years of operation, and operation stabilization. Secondly, Dakai reservoir dam was built for a long time, and it does not have the condition of continuous data monitoring and real-time transmission back, and the collection of monitoring data mainly relies on manual readings to complete, and the data collection is carried out by the operation management personnel holding the readout meter at a certain period of time. In addition, the dam monitoring data belongs to the internal information of the project operation and management unit, we do not have the authority to disclose all the monitoring data, and the measured values appearing in this paper are also disclosed with the consent of the operation and management unit. In order to solve your concerns and troubles, we have added pictures of the monitoring equipment and data acquisition process in the uploaded revised manuscript with track changes (3.4 Monitoring System Layout), and explained the process of monitoring data acquisition, which hopefully can answer your question.

5. Regarding the incomplete discussion section: In response to your suggestion, we have expanded the discussion section, please see lines 515-561 in the revised manuscript with track changes.

We sincerely hope that these changes can solve the problems you raised and improve the quality of the article, we are very grateful for your valuable comments and look forward to your further feedback, if by any other need to improve the place, we are willing to continue to strive to improve.

Reviewer #5: 

The quality of this manuscript has been relatively high after two rounds of revisions. And it is to be recognised and appreciated that the innovativeness of this manuscript is very clear, as the authors make it clear in the introduction that existing studies have ignored the effects of rheology. This study proposes a new combined rheological cell model based on the classical combined cell model and also simulates the seepage and displacement of a deep soft soil composite dam foundation using COMSOL software. However I think this manuscript should also address the following minor issues before acceptance:

1.It is suggested that the formula of 122 lines can be numbered.

Response: In response to your suggestion, we have numbered the formulas in line 122 and reordered and renumbered all the formulas that follow, as shown in lines 122-159 of the revised manuscript with track changes.

2.On line 151, 'see in the picture' should be 'see in Fig.3'

Response: In accordance with your suggestion, we have changed “the picture” to “Fig 3” to make the expression more accurate, as shown in line 151 of the revised manuscript with track changes.

3. The "Kpa" in the text should be "kPa", for example, line 187, line 190.

Response: We have checked all the international units in the manuscript and corrected the incorrect writing, and we thank you for these details that helped us avoid unnecessary errors, as shown in lines 190 and 192 of the revised manuscript with track changes.

4. The author is requested to add more detailed information of the project example in "3.1 Summary of project", such as the location of the dam in China, etc. A special picture needs to be drawn, which should show a complete map of China and the location or coordinates of the project.

Response: In response to your suggestion, we have added a picture of the project location in 3.1 Dam Overview, which clearly shows where Dakai reservoir is located in China, as shown on line 174 of the revised manuscript with track changes.

5. In "3.4 Monitoring system layout" of the manuscript, the author shows various kinds of sensors that can be used to monitor the displacement and seepage of DAMS. I am very curious about whether the author installed these sensors 11 years ago, dug up the dam and buried the sensors in it, or how they were installed inside the rock and soil body. It is well known that it is difficult to bury sensors into the interior of a geotechnical body, but the authors seem to have done it easily and buried many different types of sensors. And I'm also wondering, did the author just put the computer and collector directly on the site for 11 years, which doesn't seem realistic, or did he utilise some kind of wireless networked data transmission device to monitor the site remotely? None of this key information and details are mentioned by the authors in the latest version of the manuscript. The author does not describe in detail when the monitoring time started and ended, and the text does not contain any pictures of on-site installation of sensors and equipment. I don't think it's appropriate.

Response: First of all, the clay core wall rockfill dam selected in this paper has been designed with special monitoring design, based on the relevant specifications approved and released by the Ministry of Water Resources of China, according to the project grade, scale, structural type and its topography, geological conditions and geography, etc., to set up the necessary monitoring projects and their corresponding facilities, the specific monitoring system layout and the equipment used are described in the 3.4 part of the manuscript. During the construction of the dam, the installation of the monitoring equipment was carried out as part of the construction process. As the dam is filled, the monitoring equipment is installed inside the dam, and then the signal lines and cables of the monitoring equipment are centrally led out to the observation station at the top of the dam, where subsequent monitoring is carried out. In addition, the monitoring equipment will be protected accordingly before the layering of the dam, which is used to ensure the safety of the equipment. When the monitoring equipment is installed and debugged correctly, the data collection of the displacement and deformation of the dam body, stress and strain, etc., can be carried out, and the monitoring process will continue with the operation of the dam. At present, the equipment and construction process for dam monitoring in China are quite mature, and the monitoring instruments have been developed to the stage of automatically collecting readings and transmitting them in real time, whereas Dakai Reservoir Dam adopts the form of manually collecting readings at regular intervals, and compared with the automated monitoring technology, the traditional and conventional monitoring methods do not have significant innovations, therefore, we did not introduce this part of the manuscript intentionally. Section 3.4 of the revised manuscript with track changes provides an explanation of the monitoring data collection process, which we hope will enable the reader to better understand.

6. According to the description in this paper, these devices are quite advanced, the longest monitoring time can reach 4000 days, the equipment and sensors run continuously for more than 10 years, and the data acquisition instruments are constantly powered up during operation, which is very scary, and I am a little curious about how the author did it. In order to let readers have a better understanding of the equipment, it is best to add a special section detailing the content of the equipment and devices, introducing the models of the equipment and sensors used in the field test of this study, and it is best to put the schematic diagram of the equipment.

Response: As you can see, China has now developed to the stage of intelligent and full life cycle monitoring in dam monitoring, where the monitoring equipment will carry out continuous monitoring work with the operation of the dam. The monitoring equipment in this manuscript does not need to be energized for a long period of time after installation. When we need to collect monitoring data at a certain stage or a certain time, we only need to connect the data collection equipment to the signal line of the monitoring equipment, and readings can be completed through the transmission of signals. Since the monitoring equipment and methods used in this project are not state-of-the-art at present, and the focus of this manuscript is to verify the feasibility of the established rheological coupling model in simulating soft soil composite foundations, there is no dedicated section in the manuscript to introduce the monitoring system of the dam. According to your suggestion, in order to provide the readers with a better understanding, we have explained the process of monitoring data acquisition in 3.4 Monitoring System Layout and added pictures and flowcharts of the monitoring equipment, which are shown in lines 239-250 of the revised manuscript with track changes. The following figure shows a schematic diagram of how the earth pressure gauge works.

Reviewer #7: 

This paper establishes a new rheological model of soft soil composite foundations to study the deformation, stress and rheological characteristics of core wall rockfill dam and composite foundations. This topic is interesting and important for engineering practice. However, there are some outstanding issues that must be noted:

1. The papers writing logic is poor and fails to grasp the core description. Keywords are not accurate enough. For example, why is “elemental modeling” used as a keyword? It is suggested that the author should strengthen the revision of paper writing.

Response: You pointed out that our use of “elemental modeling” as a keyword is not accurate enough, we totally agree with you, so we have reevaluated and chosen a more accurate keyword to reflect the research content and focus of the paper. As for the problem of illogicality in the thesis, we did find that the section on numerical modeling was a bit confusing, so we have rearranged and reorganized the logic of the confusing chapters, and split the original Chapter 3 (Analysis of engineering examples) into two chapters, namely Chapter 3 (Engineering overview) and Chapter 4 (Numerical modeling). This change is more in line with the steps of numerical modeling and makes the text more clearly organized. Please see the document “Revised Manuscript with Track Changes” for details.

2. The description of “Soft soil composite foundation rheological modeling” in Part 2 is confused. For example, E2 in formula 4 and S in formula 6 are not explained, and Eα in the later explanation of formula 7 does not appear in the formula, so I do not know what it means.

Response: Thank you very much for pointing out the details in the manuscript. In fact, during the last revision, we have found that some formulas were written with err

---

## [Decision Letter · Decision Letter 3]

PONE-D-24-13121R3Research on composite foundation rheology using the combined rheological element modelPLOS ONE

Dear Dr. MAO,

Thank you for submitting your manuscript to PLOS ONE. After careful consideration, we feel that it has merit but does not fully meet PLOS ONE’s publication criteria as it currently stands. Therefore, we invite you to submit a revised version of the manuscript that addresses the points raised during the review process.

We look forward to receiving your revised manuscript.

Kind regards,

Shamshad Alam, PhD

Academic Editor

PLOS ONE

Journal Requirements:

Reviewers' comments:

Reviewer's Responses to Questions

**Comments to the Author**

1. If the authors have adequately addressed your comments raised in a previous round of review and you feel that this manuscript is now acceptable for publication, you may indicate that here to bypass the “Comments to the Author” section, enter your conflict of interest statement in the “Confidential to Editor” section, and submit your "Accept" recommendation.

Reviewer #3: All comments have been addressed

Reviewer #8: (No Response)

2. Is the manuscript technically sound, and do the data support the conclusions?

Reviewer #3: Yes

Reviewer #8: Yes

3. Has the statistical analysis been performed appropriately and rigorously? 

Reviewer #3: Yes

Reviewer #8: Yes

4. Have the authors made all data underlying the findings in their manuscript fully available?

Reviewer #3: Yes

Reviewer #8: Yes

5. Is the manuscript presented in an intelligible fashion and written in standard English?

Reviewer #3: Yes

Reviewer #8: Yes

6. Review Comments to the Author

Reviewer #3: After several rounds of revisions, the quality of the article has been slightly improved, but there are still some issues. For example, regarding Figure 4, I strongly suggest that the authors refer to more published papers on how to present maps. They can use software such as Google Earth or ArcGIS to present the map of China. Currently, there are a lot of Chinese characters on the figure. It should be noted that this is an international journal. In terms of the elaboration of innovativeness, the comparison with cutting-edge research is not in-depth enough, and the advantages of the new model in international similar research have not been fully highlighted. In addition, there are problems such as poor consistency in the use of professional terms, some expressions that are not concise enough, and incomplete information labeling in the figures, as well as issues with the clarity and aesthetics of the figures that need to be improved.

Reviewer #8: The paper proposes a novel combined rheological element model based on classical composite element modeling and applies it to simulate the rheology-consolidation coupling behavior of deep soft soil composite dam foundations. And the authors have made explicit changes and improvements to the paperin response to the reviewers' comments. However, the following improvements are suggested:

1、The transition between Section 4.3.2 ("Parameter Verification") and Section 5 ("Results Analysis") is abrupt. A brief overview outlining the objectives and framework of subsequent analyses should be added before Section 5.1.

2、Contour labels in Figures 13–16 are unclear, adjusting color contrast or adding zoomed-in insets would improve readability. And time axes in Figures 9 and 12 are labeled as "Time(d)" without clarifying "days."

3、The conclusion mentions "scientifically considering rheological effects" but lacks specific engineering measures, practical guidance based on model results should be added.

7. PLOS authors have the option to publish the peer review history of their article (what does this mean?). If published, this will include your full peer review and any attached files.

Reviewer #3: No

Reviewer #8: No

---

## [Author Response · Author response to Decision Letter 4]

23 Mar 2025

Replies to the reviewers’ comments:

Reviewer #3:

After several rounds of revisions, the quality of the article has been slightly improved, but there are still some issues. For example, regarding Figure 4, I strongly suggest that the authors refer to more published papers on how to present maps. They can use software such as Google Earth or ArcGIS to present the map of China. Currently, there are a lot of Chinese characters on the figure. It should be noted that this is an international journal. In terms of the elaboration of innovativeness, the comparison with cutting-edge research is not in-depth enough, and the advantages of the new model in international similar research have not been fully highlighted. In addition, there are problems such as poor consistency in the use of professional terms, some expressions that are not concise enough, and incomplete information labeling in the figures, as well as issues with the clarity and aesthetics of the figures that need to be improved.

Response: First of all, I would like to express my sincere appreciation for the time and effort you have invested in reviewing our manuscript. We value the valuable comments you have provided and have carefully considered each piece of feedback.

1.Regarding the issue that Figure 4 does not meet the national journal standards:

Thank you for your constructive suggestions regarding the presentation of Figure 4. We have carefully revised the map according to your feedback: 1) Language Standardization: All Chinese characters in the original figure have been removed and replaced with English labels to align with the international journal standards. 2) Cartographic Compliance: The revised map was regenerated using ArcGIS with strict adherence to China’s mapping regulations and international publishing guidelines.

The updated Figure 4 has been submitted with the revised manuscript. We confirm that all maps comply with both Chinese legal standards and the journal’s cartographic policies. Your attention to detail has significantly improved the quality of our work.

2.Regarding the lack of clarity in innovative articulation: Thank you for your valuable input on highlighting the innovative and international relevance of our work. In the revised manuscript, we strengthen the discussion of rheological models and explicitly address their limitations in terms of long-term settlement prediction and parameter identification challenges. In order to reduce the gap between the existing model and the actual engineering requirements, we propose a hierarchical modeling method with clear mathematical concepts and simple parameter calibration. Unlike previous studies which only focused on consolidation or rheology, our model not only characterized the consolidation and rheology behavior of soft soil, but also considered the seepage and stress coupling of the foundation, which is crucial for the study of deep soft clay foundation. The advantages of this model are verified by the case study of Dakai clay core rockfill dam. Compared with the traditional consolidation model, the settlement prediction error of this model decreases gradually with time and is much lower than that of the consolidation model. We believe that these revisions fully demonstrate the originality and technological advancement of our approach. The above changes have been highlighted in yellow in the introduction to the revised manuscript, specifically on lines 26-29, 53-60 and 69-87.

3. Regarding the poor use of technical terms and the lack of concise language: We sincerely appreciate your valuable feedback on the standardization of terms and the simplicity of language. In response to your comments, we have made systematic improvements throughout the manuscript:

1)Unity of terminology

To ensure professional coherence, we have systematically standardized terminology throughout the manuscript:

①Fluid-solid interaction: Replaced non-standard terms (e.g., "fluid-structure"→"fluid-solid"; "underlyment"→"underlying stratum").

②Engineering terms: Revised structural nomenclature (e.g., "clay core wall rockfill dam"→"clay core rockfill dam"; "core wall"→"clay core"; "repressive platforms"→"toe berm").

③Unit standardization: Abbreviated units in compliance with SI conventions (e.g., "meters"→"m").

④Technical vocabulary: Optimized key terms (e.g., "gravel piles"→"stone columns"; "osmometer"→"piezometer"; "seepage control wall"→"cutoff wall"; "reinforcing layer"→"reinforced zone").

All terminology revisions are highlighted in green in the revised manuscript.

2�Expression Conciseness

We have rigorously streamlined redundant descriptions to enhance clarity:

①Section 2.3 (Lines 158–161) & Section 3.3 (Lines 198–208): Condensed repetitive explanations while preserving technical accuracy.

②Dam description (Lines 176–179): Removed redundant qualifiers and restructured sentences for improved readability.

Optimized sections are marked with light blue highlighting for ease of review.

We believe these revisions have substantially enhanced the manuscript's professionalism and conciseness. Should any further refinements be required, we welcome the reviewer's guidance. Thank you once again for your invaluable feedback.

4.Regarding the issue of image resolution: we have meticulously reviewed all figures in the manuscript and implemented the following revisions: 1)Re-annotated textual elements within the figures (e.g., Figure 6) to ensure clarity and consistency. 2)Updated the annotations in relevant figures (Figures 5 and 7) following the standardization of terminology throughout the text. 3)Addressed instances of unclear numerical representations in Figures 13 to 16 by enhancing their visibility and legibility.

Reviewer #8: 

The paper proposes a novel combined rheological element model based on classical composite element modeling and applies it to simulate the rheology-consolidation coupling behavior of deep soft soil composite dam foundations. And the authors have made explicit changes and improvements to the paperin response to the reviewers' comments. However, the following improvements are suggested:

1、The transition between Section 4.3.2 ("Parameter Verification") and Section 5 ("Results Analysis") is abrupt. A brief overview outlining the objectives and framework of subsequent analyses should be added before Section 5.1.

Response:

We sincerely appreciate the reviewer’s insightful comment. In response to the suggestion, we have added a transitional paragraph at the beginning of Section 5 ("Results Analysis"), prior to Section 5.1, to explicitly outline the objectives and framework of the subsequent analyses. The added text includes:

1) A reaffirmation of the model’s reliability based on validation results, emphasizing the consistency between simulations and measurements.

2)A clear statement that this section will systematically investigate the time-dependent evolution of multi-physical fields in the composite foundation under three working conditions, focusing on deformation, stress, and seepage.

3)A declaration of the intent to identify critical thresholds governing long-term stability through quantitative analysis.

This change is intended to enhance the logical connection between chapters and to help the reader grasp the core direction of the subsequent analysis in advance. The added text has been indicated in blue in the revised manuscript, as shown in lines 414-420. We thank the reviewer for their meticulous evaluation and constructive feedback.

2、Contour labels in Figures 13–16 are unclear, adjusting color contrast or adding zoomed-in insets would improve readability. And time axes in Figures 9 and 12 are labeled as "Time(d)" without clarifying "days."

Response:

We appreciate this constructive feedback. The following revisions have been implemented:

1)Label clarity: Enlarge the contour labels in Figures 13 to 16 and adjust its font size according to the background so that readers can see it clearly.

2)Unit ambiguity: We have revised the time axis labels in Figures 9 to 12 to explicitly state the unit as “Time (days)”. Additionally, all units in the manuscript have been standardized to avoid similar ambiguities.

3、The conclusion mentions "scientifically considering rheological effects" but lacks specific engineering measures, practical guidance based on model results should be added.

Response:

We sincerely thank the reviewer for their valuable feedback. In response to the suggestion to include specific engineering measures in the conclusion, we have added a detailed discussion of engineering control methods and practical guidance based on the model results in the final paragraph of Section 6.1. The added measures include:

1) During the foundation treatment phase: extending the pre-loading period of the dam foundation to ensure adequate consolidation of soft soil; installing a three-dimensional drainage system (with longitudinal and transverse configurations) to accelerate the dissipation of pore water pressure; increasing the penetration depth of reinforcement piles to enhance bearing capacity.

2) During the dam filling phase: adopting intermittent staged loading to mitigate rheological risks.

3) During the dam operation phase: implementing long-term settlement monitoring to ensure operational safety.

These measures are directly linked to the model-derived insights on pore pressure evolution, stress redistribution, and time-dependent deformation patterns. The revisions strengthen the translation of theoretical findings into practical engineering applications. The reviewer's suggestion has significantly enhanced the practical value of the conclusions, and we are deeply grateful for their constructive input. Please refer to the red text in the revised manuscript, particularly lines 573–580, for the specific changes.

---

## [Decision Letter · Decision Letter 4]

PONE-D-24-13121R4Research on composite foundation rheology using the combined rheological element modelPLOS ONE

Dear Dr. MAO,

Thank you for submitting your manuscript to PLOS ONE. After careful consideration, we feel that it has merit but does not fully meet PLOS ONE’s publication criteria as it currently stands. Therefore, we invite you to submit a revised version of the manuscript that addresses the points raised during the review process.

We look forward to receiving your revised manuscript.

Kind regards,

Shamshad Alam, PhD

Academic Editor

PLOS ONE

Journal Requirements:

Reviewers' comments:

Reviewer's Responses to Questions

**Comments to the Author**

1. If the authors have adequately addressed your comments raised in a previous round of review and you feel that this manuscript is now acceptable for publication, you may indicate that here to bypass the “Comments to the Author” section, enter your conflict of interest statement in the “Confidential to Editor” section, and submit your "Accept" recommendation.

Reviewer #3: All comments have been addressed

Reviewer #5: All comments have been addressed

Reviewer #9: (No Response)

2. Is the manuscript technically sound, and do the data support the conclusions?

Reviewer #3: Yes

Reviewer #5: Yes

Reviewer #9: No

3. Has the statistical analysis been performed appropriately and rigorously? 

Reviewer #3: Yes

Reviewer #5: I Don't Know

Reviewer #9: No

4. Have the authors made all data underlying the findings in their manuscript fully available?

Reviewer #3: Yes

Reviewer #5: Yes

Reviewer #9: No

5. Is the manuscript presented in an intelligible fashion and written in standard English?

Reviewer #3: Yes

Reviewer #5: Yes

Reviewer #9: No

6. Review Comments to the Author

Reviewer #3: The authors have basically addressed the issues and concerns raised by the reviewers. The decision on acceptance is left to the editors.

Reviewer #5: The manuscript has undergone four rounds of revisions and now demonstrates substantial improvement in clarity, technical rigor, and presentation. The proposed combined rheological model effectively addresses the rheology-consolidation-seepage coupling in deep soft soil composite foundations, and the case study validation using COMSOL is well-executed. While the paper is nearly ready for acceptance, minor revisions are required to address lingering issues related to clarity, methodological details, and contextualization within the broader literature.

I believe that once this manuscript is accepted and published, it will be cited by other scholars with a high probability and frequency. I hope the author can continue to carry forward the scientific research spirit of not giving up, not giving up and working hard, and implement the final revision well. The following are some final revision suggestions I put forward:

1. The literature review in the introduction (Lines 24–79) focuses on consolidation and rheological models but lacks explicit references to recent studies on deformation monitoring and seepage analysis in similar geotechnical contexts. For instance, the following works could enrich the discussion:

- https://doi.org/10.1038/s41598-024-57598-7

- https://doi.org/10.1007/s10064-024-03793-9

- https://doi.org/10.1007/s10040-024-02844-5

These academic literatures align with the manuscript’s focus on time-dependent deformation and seepage-stress coupling, and their inclusion would better situate the novelty of the proposed model.

2. Section 3.4: Clarify Monitoring Equipment Specifications. The monitoring system (Lines 222–258) mentions instruments like "electromagnetic settlement inclinometer tubes" and "piezometers" but omits critical details such as manufacturer names, model numbers, or measurement accuracies.

For example:

Original text (Lines 229–234): "A mobile inclinometer measures the horizontal displacement..."

Revision suggestion: "Specify the inclinometer model (e.g., *SINCO Digitilt*) and its resolution (e.g., ±0.1 mm/m). "

3. Section 4.4: The numerical modeling section (Lines 327–350) describes mesh generation (Figure 10) but does not validate grid independence. A brief sensitivity analysis (e.g., comparing results for coarse vs. fine meshes) should be added to ensure computational accuracy.

4. Stress values in Table 7 (Page 34) are reported in MPa, but the text (Line 511) refers to "0.00075mpa/month." Standardize units to either MPa or kPa throughout the manuscript. Authors are asked to read through and check the manuscript carefully to ensure that the units of all parameters are consistent and standardized.

7. There are problems with the grammar or sentence structure of some sentences in the manuscript, and I would suggest that the author check the entire text and touch up the language in the body.

For example:

Line 446: "impounding period's growth rate is significantly larger than the impoundment period's..." → "the growth rate during impoundment is significantly higher...".

Line 608: "since it will have an impact" → "as it may adversely affect."

Reviewer #9: DECLINED to be recommended. A draft after 4 rounds of revision doesn’t have keyword???

Fig 9 denotes the modeling flowchart?? It trivially describes the time dependency, while the draft doesn’t provide how time analysis of data is steps of filling were analyzed!!! On the other hand, data are time dependent and require continuous updating (dynamic database) while this work doesn’t provide a solution for uniformly normalizing dynamic databases via different factors. Search for new approaches via keywords like normalizing large scale sensor-based data with an automated method …

Lack of convincing information about the used assumptions? Type of mesh and why? The boundary condition and its definition? Used data for model construction and the method of incorporation?

Considering the SSI???

The given FEM model definitely is incorporated with truncation and round off errors. How did you evaluate these errors and with what criteria??

COMSOL only analyzes the nonlinearity of the model, not the actual structure. Since the response of a structure is sensitive to the strengths and stiffnesses of its components, the actual properties may not be known accurately. Capacity design can greatly reduce uncertainty. No information regarding these problems and treatment methods has been given.

Discussion should be documented via technical limitations/physical interpretation of the results/impact of bias of the used data on the results/real practical and evidential analysis/considering the effect of subsurface spatial distribution of soil/rock types; the effect of clay sensitivity; involved uncertainty for the post processing geo-model (may get good hints from https://link.springer.com/article/10.1007/s10064-018-1400-9,
https://www.sciencedirect.com/science/article/abs/pii/S0013795224002655,
https://www.mdpi.com/2220-9964/10/5/341,
https://link.springer.com/article/10.1007/s10064-018-1400-9,
https://www.sciencedirect.com/science/article/abs/pii/S0013795215000411,
https://www.sciencedirect.com/science/article/pii/S0341816222002752,
https://link.springer.com/article/10.1007/s00366-023-01852-5, …)/computational time and cost/solid comparison with other scholars and models…

English revision with the help of a native agent

Verification of the model is not certified. It doesn’t provide solid comparative analysis

Improper captions. For example, Figs like 11 have different parts but none of them are reflected in the caption

7. PLOS authors have the option to publish the peer review history of their article (what does this mean?). If published, this will include your full peer review and any attached files.

Reviewer #3: No

Reviewer #5: No

Reviewer #9: No

---

## [Author Response · Author response to Decision Letter 5]

5 Jun 2025

Dear academic editor and reviewers:

Thank you for your letter and the reviews’ comments on our manuscript entitled “Research on composite foundation rheology using the combined rheological element model ” (PONE-D-24-13121).Those comments are very helpful for revising and improving our paper, as well as the important guiding significance to other research. We have studied the comments carefully and made corrections which we hope meet with approval. The main corrections are in the revised manuscript and the responds to the reviews’ comments are as follows.

Replies to the reviewers’ comments:

Reviewer #3:

The authors have basically addressed the issues and concerns raised by the reviewers. The decision on acceptance is left to the editors.

Response: Thank you for your constructive feedback and confirmation that our revisions have adequately addressed the concerns raised. We appreciate your professional assessment of our work.

Reviewer #5: 

The manuscript has undergone four rounds of revisions and now demonstrates substantial improvement in clarity, technical rigor, and presentation. The proposed combined rheological model effectively addresses the rheology-consolidation-seepage coupling in deep soft soil composite foundations, and the case study validation using COMSOL is well-executed. While the paper is nearly ready for acceptance, minor revisions are required to address lingering issues related to clarity, methodological details, and contextualization within the broader literature.

I believe that once this manuscript is accepted and published, it will be cited by other scholars with a high probability and frequency. I hope the author can continue to carry forward the scientific research spirit of not giving up, not giving up and working hard, and implement the final revision well. The following are some final revision suggestions I put forward:

1. The literature review in the introduction (Lines 24–79) focuses on consolidation and rheological models but lacks explicit references to recent studies on deformation monitoring and seepage analysis in similar geotechnical contexts. For instance, the following works could enrich the discussion:

- https://doi.org/10.1038/s41598-024-57598-7

- https://doi.org/10.1007/s10064-024-03793-9

- https://doi.org/10.1007/s10040-024-02844-5

These academic literatures align with the manuscript’s focus on time-dependent deformation and seepage-stress coupling, and their inclusion would better situate the novelty of the proposed model.

Response: We sincerely appreciate your suggestion to strengthen the literature review with recent advances in deformation monitoring and seepage analysis. As implemented in the revised introduction (Lines 80-86), we have incorporated new references addressing integrated field monitoring and coupled analysis [29,30]. Furthermore, we have also revised the first paragraph of the introduction, emphasizing the impact of rheology on the safety of the dam. For details, please refer to lines 36-42 of the revised manuscript.

2. Section 3.4: Clarify Monitoring Equipment Specifications. The monitoring system (Lines 222–258) mentions instruments like "electromagnetic settlement inclinometer tubes" and "piezometers" but omits critical details such as manufacturer names, model numbers, or measurement accuracies.

For example:

Original text (Lines 229–234): "A mobile inclinometer measures the horizontal displacement..."

Revision suggestion: "Specify the inclinometer model (e.g., *SINCO Digitilt*) and its resolution (e.g., ±0.1 mm/m). "

Response: Thank you for your valuable comments. We have carefully revised the manuscript to clarify the monitoring equipment specifications according to your suggestions. In Section 3.4 of the manuscript, for instruments such as electromagnetic settlement inclinometers, portable inclinometers, fixed inclinometers, soil displacement meters, earth pressure transducers and piezometers, the manufacturer models and resolutions have been added. For details, see the yellow highlighted text in lines 232-247. Meanwhile, we have also modified the grammar and structure of the sentences in Section 3.4.

These revisions have enhanced the repeatability of the monitoring system design and enabled better evaluation of measurement accuracy. We believe that these details are now sufficient to address the technical requirements for describing engineering monitoring instruments.

3. Section 4.4: The numerical modeling section (Lines 327–350) describes mesh generation (Figure 10) but does not validate grid independence. A brief sensitivity analysis (e.g., comparing results for coarse vs. fine meshes) should be added to ensure computational accuracy.

Response: We sincerely thank the reviewers for their valuable suggestions on the verification of grid independence. To solve this problem, we added a comprehensive grid sensitivity analysis in Section 4.4 (lines 348-361) and made the following key enhancements:

1. Comparison of meshes with different resolutions: The model is meshed into three levels: coarse, medium, and fine, and the key areas are refined with emphasis, as shown in Figure 10.

2. Quantitative parameter comparison: Three safety-critical parameters under grid resolution were evaluated, as shown in Table 5;

3. Comparison of computational efficiency: The computational times of the three grids were compared, as shown in Table 5.

Ultimately, all simulations retained a medium grid (4,892 elements) because it achieved engineering accuracy while maintaining computational efficiency.

Your opinions have enhanced the rigor of the calculation and confirmed the reliability of our results. We are grateful to the reviewers for this important improvement of our research.

4. Stress values in Table 7 (Page 34) are reported in MPa, but the text (Line 511) refers to "0.00075mpa/month." Standardize units to either MPa or kPa throughout the manuscript. Authors are asked to read through and check the manuscript carefully to ensure that the units of all parameters are consistent and standardized.

Response: Thank you for your careful review and valuable suggestions. We have standardized the units throughout the manuscript as follows:

1. Unit Consistency:

All stress values in tables and the text are now expressed in kPa, MPa and GPa with SI conventions. Replace “0.00075mpa/month”with “0.00075Mpa/month”. See line 629 for details.

2. Global Verification:

We conducted a full-text check to ensure consistency of units for all parameters�e.g. change KN/m³ to kN/m³ in Table 1 and Table 2). These revisions improve the clarity and professionalism of the manuscript. Please see the highlighted changes in the revised manuscript. We appreciate your attention to this important detail.

5. There are problems with the grammar or sentence structure of some sentences in the manuscript, and I would suggest that the author check the entire text and touch up the language in the body.

For example:

Line 446: "impounding period's growth rate is significantly larger than the impoundment period's..." → "the growth rate during impoundment is significantly higher...".

Line 608: "since it will have an impact" → "as it may adversely affect."

Response:

We sincerely appreciate the reviewer's meticulous attention to linguistic precision. In response to this comment, we have conducted a comprehensive, line-by-line revision of the manuscript with the following actions:

1. Grammar Refinement

The past passive voice describing the general method tense becomes present tense (e.g. , Line 211: replaced "was adopted to..." with "dictates employing...")

2.Terminology Standardization

Ensured unified terminology across all sections (e.g., replaced "permeability coefficient" with "hydraulic conductivity")

3. Elimination of Redundancy

①Removed colloquial expressions (e.g., Line 199: replaced "that can be dug up to ..." with "exhibiting… ")

②Streamlined verbose constructions (e.g., Line 195: condensed "which is loose and highly permabele..." to "loose, highly permeable...")

4. Implementation of Reviewer's Specific Suggestions

①Line 459: Revised to "the growth rate during impoundment being significantly higher than during filling..."

②Line 623: Modified to "as it may adversely affect..."

All modifications have been marked in red in the revised manuscript. We believe these refinements significantly elevate the manuscript's linguistic quality and readiness for publication.

Reviewer #9: 

1.DECLINED to be recommended. A draft after 4 rounds of revision doesn’t have keyword???

Response: Thank you for highlighting this important point. We sincerely apologize for the oversight of not including keywords in the manuscript body.

During initial submission, we strictly followed PLOS ONE’s official formatting sample (formatting_sample_main_body), which did not contain a keywords section. However, we confirm that keywords (Rheology; Rheological and consolidation coupling model; Soft soil composite foundation; Stress deformation; Seepage) were fully entered in the designated field within the online submission system. We acknowledge that keywords should appear in the manuscript for clarity. We will immediately add the following section to the revised manuscript:

Keywords: Rheology; Rheological and consolidation coupling model; Soft soil composite foundation; Stress deformation; Seepage

This will be placed below the Abstract in accordance with journal guidelines. We appreciate your vigilance in improving our manuscript’s completeness.

2. Fig 9 denotes the modeling flowchart?? It trivially describes the time dependency, while the draft doesn’t provide how time analysis of data is steps of filling were analyzed!!! On the other hand, data are time dependent and require continuous updating (dynamic database) while this work doesn’t provide a solution for uniformly normalizing dynamic databases via different factors. Search for new approaches via keywords like normalizing large scale sensor-based data with an automated method …

Response: We sincerely appreciate the reviewer's attention to the temporal aspects of our model. It should be clarified that this study belongs to geotechnical rheological process simulation. The temporal dependency originates from physical constraints of the soft soil rheological constitutive relationship and the construction/impoundment schedule, rather than dynamic data stream processing in computer science. We have reinforced this core attribute through the following modifications:

1. The temporal workflow is strictly driven by physical timelines.

①Filling completion period: Incremental loading inherits stress states from previous stages (Prestressed Study), with timesteps defined by the construction schedule (3 months/stage).

②Initial impoundment period: Water level variation follows the physical function

h(t) = 75 + 0.5t (m) (0.5 m/day), completing impoundment in 36 days.

③Operational stabilization period: Steady-state analysis covers 10 years of physical time.

2. Physical time progression is explicitly handled by COMSOL's Time-Dependent Solver, with inter-stage field variable transfer via boundary condition inheritance (no database updating required).

①We have explicitly visualized the physical timeline in the revised Fig. 9: Added stage duration labels (e.g., 3 months)

②Annotated the impoundment function h(t)

③ Indicated physical progression direction with timeline arrows .

3. Inapplicability of Specific Terminology:

The mentioned concepts (e.g., automated normalization, dynamic data updating) are not applicable to our physics-based model.

①"Dynamic Updating" manifests as inheritance of physical field variables, not database operations.

②The core innovation lies in the composite rheological unit model, not data algorithms (see Abstract and Section 2).

These modifications further clarify that temporal dimensionality in this study is an inherent attribute of the physical process, not an independent data processing workflow. We thank the reviewer for prompting us to present this key aspect more clearly.

3. Lack of convincing information about the used assumptions? Type of mesh and why? The boundary condition and its definition? Used data for model construction and the method of incorporation?

Response: We thank the reviewer for raising these critical methodological aspects. Detailed information is already provided in our manuscript, and we clarify each point below with specific cross-references:

1. Key Assumptions:

①Unsaturated→saturated transition: Richards' equation

②Saturated foundation: Poroelastic theory

③ Material models: Composite rheology (soft soil), Duncan E-v (rockfill), Linear elasticity (cutoff wall).

2. Mesh Selection Rationale :

①Type: Locally refined triangle meshes.

②Reason: Medium mesh (4,892 elements) optimized accuracy (vs. fine mesh: <2% error in critical displacements/stresses) and computational efficiency.

3. Boundary Conditions :

①Hydraulic: Reservoir h(t)=75+0.5t (m); Foundation base: zero pressure.

②Mechanical: Foundation sides - roller supports; Base - fixed.

③Initial: Geostatic balance + staged stress inheritance.

4. Data Integration:

①Parameters from: field monitoring + inverse analysis for rheology.

②Validation: inversion parameter vs. experimental parameter (Table 3).

③Implementation: COMSOL material modules + Prestressed Study.

These elements collectively ensure the model’s fidelity to real-world dam behavior.

4. Considering the SSI???

Response: Thank you for highlighting this fundamental aspect. Soil-structure interaction (SSI) is intrinsically embedded in our modeling framework through three interconnected mechanisms. First, mechanical coupling is achieved via the proposed composite rheological unit (Section 2), which explicitly integrates dam structure and soft soil foundation behavior. This enables stress transfer at interfaces through staged loading simulation and nonlinear rockfill-soil interaction resolution (Duncan E-v model in Section 4.3.1). Second, hydromechanical coupling captures the closed-loop SSI: reservoir impoundment induces pore pressure changes → alters effective stresses → drives foundation deformation → feeds back to dam stability. This process is quantified using Richards' equation for unsaturated-saturated transition and Biot's consolidation theory (Eq. 40, Section 4.2.1) for saturated conditions. Third, long-term SSI effects manifest through continuous stress redistribution between the dam and foundation via soft soil creep, driving post-construction settlement and decadal-scale stress evolution. In essence, SSI forms the operational core of our coupled model rather than functioning as an add-on component.

5. The given FEM model definitely is incorporated with truncation and round off errors. How did you evaluate these errors and with what criteria??

Response: We appreciate this insightful technical query regarding numerical errors.

Our error management strategy aligns with standard FEM engineering practices:

1. Truncation Error Evaluation:

Spatial discretization error was quantified via mesh convergence tests (Coarse/Medium/Fine meshes in Table 5). Maximum variation in critical outputs (displacements/settlements/stresses) was <2% between mesh levels–well below the 5% threshold accepted in geotechnics.

2. Round-off Error Control:

COMSOL's default double-precision solverreduced single-operation rounding errors to 10-16. The global stiffness matrix condition number remained <104 , preventing significant error amplification.

These protocols ensure numerical uncertainties are negligible relative to the model's predictive capability .

6. COMSOL only analyzes the nonlinearity of the model, not the actual structure. Since the response of a structure is sensitive to the strengths and stiffnesses of its components, the actual properties may not be known accurately. Capacity design can greatly reduce uncertainty. No information regarding these problems and treatment methods has been given.

Response: We appreciate the reviewer's emphasis on structural nonlinearity and uncertainty management. While capacity design is essential for seismic structures, our study focuses on validating the composite rheological model under controlled geotechnical processes. To address material property uncertainties, thre

---

## [Decision Letter · Decision Letter 5]

Research on composite foundation rheology using the combined rheological element model

PONE-D-24-13121R5

Dear Dr. MAO,

We’re pleased to inform you that your manuscript has been judged scientifically suitable for publication and will be formally accepted for publication once it meets all outstanding technical requirements.

Kind regards,

Hui Yao

Academic Editor

PLOS ONE

Additional Editor Comments (optional):

Reviewers' comments:

Reviewer's Responses to Questions

**Comments to the Author**

1. If the authors have adequately addressed your comments raised in a previous round of review and you feel that this manuscript is now acceptable for publication, you may indicate that here to bypass the “Comments to the Author” section, enter your conflict of interest statement in the “Confidential to Editor” section, and submit your "Accept" recommendation.

Reviewer #3: All comments have been addressed

Reviewer #5: All comments have been addressed

Reviewer #9: (No Response)

2. Is the manuscript technically sound, and do the data support the conclusions?

Reviewer #3: Yes

Reviewer #5: Yes

Reviewer #9: Partly

3. Has the statistical analysis been performed appropriately and rigorously? 

Reviewer #3: Yes

Reviewer #5: I Don't Know

Reviewer #9: Yes

4. Have the authors made all data underlying the findings in their manuscript fully available?

Reviewer #3: Yes

Reviewer #5: Yes

Reviewer #9: No

5. Is the manuscript presented in an intelligible fashion and written in standard English?

Reviewer #3: Yes

Reviewer #5: Yes

Reviewer #9: No

6. Review Comments to the Author

Reviewer #3: All legends of cloud diagrams in the manuscript need to be supplemented with the represented physical quantities and their units.

Reviewer #5: The quality of the current manuscript is quite high. I believe the authors have made great efforts to address and respond to the suggestions or comments raised by the reviewers. I recommend that the editorial department accept this manuscript. The authors have gone through a lot. This is the fifth round of revision, and they have demonstrated extremely high professional standards and a spirit of academic dedication.

Reviewer #9: The observed inconsistencies imply another round of minor revision. Furthermore, the DOI could be provided as Link. None of the references highlighted in L766-776 have the correct and standard identifiers.

They MUST be:

44. Ghaderi A , Shahri AA , Larsson S. An artificial neural network based model to predict spatial soil type distribution using piezocone penetration test data (CPTu). Bulletin of Engineering Geology and the Environment, 2019, 78, 4579–4588, https://doi.org/10.1007/s10064-018-1400-9.

45. Ghaderi A , Shahri AA, Larsson S .A visualized hybrid intelligent model to delineate Swedish fine-grained soil layers using clay sensitivity. Catena, 214, 106289, https://doi.org/10.1016/j.catena.2022.106289

46. Abbaszadeh Shahri A, Kheiri A, Hamzeh A. Subsurface topographic modeling using geospatial and data driven algorithm. ISPRS International Journal of Geo-Information, 2021, 10(5), 341, https://doi.org/10.3390/ijgi10050341

47. Shahri AA, Shan C, Larsson S. A hybrid ensemble-based automated deep learning approach to generate 3D geo-models and uncertainty analysis[J].Engineering with Computers, 2024, 40, 1501–1516, https://doi.org/10.1007/s00366-023-01852-5

7. PLOS authors have the option to publish the peer review history of their article (what does this mean?). If published, this will include your full peer review and any attached files.

Reviewer #3: No

Reviewer #5: No

Reviewer #9: No

---

## [Editor Report · Acceptance letter]

PONE-D-24-13121R5

PLOS ONE

Dear Dr. MAO,

I'm pleased to inform you that your manuscript has been deemed suitable for publication in PLOS ONE. Congratulations! Your manuscript is now being handed over to our production team.

Kind regards,

on behalf of

Dr. Hui Yao

Academic Editor

PLOS ONE